# Biallelic *VARS* variants cause developmental encephalopathy with microcephaly that is recapitulated in *vars* knockout zebrafish

Aleksandra Siekierska, Hannah Stamberger et al.[#]

Aminoacyl tRNA synthetases (ARSs) link specific amino acids with their cognate transfer RNAs in a critical early step of protein translation. Mutations in ARSs have emerged as a cause of recessive, often complex neurological disease traits. Here we report an allelic series consisting of seven novel and two previously reported biallelic variants in valyl-tRNA synthetase (*VARS*) in ten patients with a developmental encephalopathy with microcephaly, often associated with early-onset epilepsy. In silico, in vitro, and yeast complementation assays demonstrate that the underlying pathomechanism of these mutations is most likely a loss of protein function. Zebrafish modeling accurately recapitulated some of the key neurological disease traits. These results provide both genetic and biological insights into neurodevelopmental disease and pave the way for further in-depth research on ARS related recessive disorders and precision therapies.

Aminoacyl tRNA synthetases (ARSs) play a key role in protein translation as they catalyze the attachment of specific amino acids to their cognate transfer RNA (tRNA) molecules[1,2]. The nuclear encoded *ARS* gene loci are subdivided into 17 cytoplasmic, 17 mitochondrial, and three bi-functional ARSs[3,4]. The canonical aminoacylation and proof-reading functions of ARSs are highly conserved across species. In addition, during evolution many ARSs acquired additional domains with unique structural characteristics that are not essential for tRNA charging but account for non-canonical functions[5,6]. These alternative functions are critical for cellular homeostasis and include among others: regulation of signal transduction and cell migration, angiogenesis and tumorigenesis, inflammatory responses, and control of cell death[5]. This functional diversity may in part account for the association between mutations in *ARS* genes and a broad range of human disorders, including neurological disorders, cancer, and auto-immune diseases[2].

Both monoallelic and biallelic pathogenic variants in *ARS* genes, encoding dominant and recessive disease traits, respectively, have been increasingly reported in patients with various disorders that often have predominantly neurological features. Dominant heterozygous mutations in *ARS* genes have been identified in patients with Charcot-Marie-Tooth disease and related peripheral neuropathies, including *AARS*[7], *GARS*[8], *HARS*[9,10], *MARS*[11,12], *WARS*[13], and *YARS*[12,14]. Recessive mutations have been identified in complex disorders often involving the central nervous system such as hypomyelination with brainstem and spinal cord involvement (*DARS*)[15], leukodystrophy (*RARS*)[16], congenital visual impairment and progressive microcephaly (*KARS*)[17], developmental delay with progressive microcephaly and intractable seizures (*QARS*)[18,19] and early onset epileptic encephalopathy with myelination defect (*AARS*)[20]. Interestingly, some *ARS* genes have been associated with both dominant and recessive disease traits including mutations in *AARS*[7,20], *KARS*[21], and *YARS*[14,22].

In this study, we report five newly diagnosed families with biallelic variants in valyl-tRNA synthetase (*VARS*), including seven novel *VARS* variants. In addition, we present an in-depth description of two families previously reported in a large study on brain malformations in mainly consanguineous families wherein *VARS* was reported as a candidate disease gene[23]. In vitro studies with patient-derived cell lines, including enzymatic assays, and yeast complementation assays show that recessive *VARS* mutations most likely lead to a loss-of-protein function, i.e. loss of function (LoF) alleles. A *vars* knockout (KO) zebrafish model further demonstrates that deficiency of *vars* results in microcephaly and epileptiform activity, replicating key characteristics of the human disease.

## Results

### Biallelic *VARS* variants cause developmental encephalopathy.
In total, ten patients from seven families with biallelic *VARS* variants were identified (Fig. 1a)[23]. All families were included through international collaborations or via the program Gene-Matcher[24]. All patients had global developmental delay (DD), which was already present in the first months of life in most patients, and prior to seizure onset or unrelated to epilepsy in five patients. All patients at a sufficient age for IQ testing had severe or profound intellectual disability (ID) and were nonverbal. Only two of the nine patients who had reached the walking age were able to walk independently, though both acquired this skill only at later age.

Eight out of ten patients had epileptic seizures, with onset during the neonatal or infantile period in seven patients (mean: 6

mo, median 4.3 mo). Seizure types included generalized or bilateral tonic-clonic seizures (seven patients), myoclonia (four patients), tonic seizures (one patient), focal seizures (two patients), and atypical absences (one patient). In patient 2, migrating focal seizures were documented on EEG. In four patients more than two anti-epileptic drug regimens failed meeting the definition of drug resistance[25]. No seizures were observed in patients 4 and 5 (family III), and repeated EEGs were normal. Both siblings were reported to have a notably happy demeanor resembling Angelman syndrome, but genetic testing for this syndrome was negative.

Other clinical neurological features included (axial) hypotonia (four patients), spasticity (five patients), and an ataxic gait (two patients). Three patients were reported to have significant sleep problems. Brain imaging showed cerebral atrophy in eight patients and atrophy or partial agenesis of the corpus callosum in four patients. Furthermore, hypomyelination or delayed myelination was reported in four patients. All patients had a severe, progressive microcephaly on the background of a more general failure-to-thrive. Patients 9 and 10 (carrying the same *VARS* variant) had additional systemic features including post-natal anemia and hepatosplenomegaly in patient 9. Patient 10 died at the age of 3 years as a consequence of septic shock with no history of immunodeficiency. A detailed summary of the clinical findings in all patients is provided in Supplementary Data 1.

All patients were referred for genetic testing with WES or whole-genome sequencing (WGS) after standard diagnostic work-up did not reveal a cause for their neurodevelopmental disorder (details available in Supplementary Notes 1 to 6). Three compound heterozygous *VARS* variants were identified in the affected members of families I (p.Leu434Val/p.Gly822Ser), II (p.Gln400Pro/p.Arg442Gln), and III (p.Leu78Argfs*35/p.Arg942Gln). Of note, in addition to the biallelic *VARS* variant, patient 3 also carried a rare de novo variant of uncertain significance in the brain expressed gene *UBE2O* which has been reported 1 time in the Exome Aggregation Consortium (ExAC)[26] (OMIM 617649, Supplementary Note 2). Two homozygous missense variants identified in families IV (p.Leu885Phe) and V (p.Arg1058Gln) were previously reported in a large study on brain malformations in mainly consanguineous families[23] and in families VI (Jordan ancestry) and VII (Israel/Arabic ancestry), the same, homozygous, p.Arg404Trp variant was identified. All *VARS* variants were deemed pathogenic based on their absence or presence in very low frequency in ExAC and the Genome Aggregation Database (gnomAD)[26], impact on protein level, different prediction tools, and segregation (Supplementary Table 1). All identified variants were located in the catalytic or the anticodon-binding domain of the protein with the exception of the frameshift variant, which was not located in a specific domain (Fig. 1b).

**In silico modeling in *T. thermophilus* ValRS.** To gain insights into the potential structural and functional consequences of the human missense variants, the sequence of the human enzyme was aligned to ValRS sequences from various taxa (Supplementary Figure 1), including the *T. Thermophilus* ValRS sequence, whose structure has been determined in complex with its cognate tRNA[27,28]. Of the eight *VARS* missense variants identified, seven could be mapped on to the corresponding residues in *T. Thermophilus* ValRS, allowing inferences to be drawn regarding their impact on protein structure and/or substrate interaction (Fig. 1c, d and Supplementary Table 2). Employing this analysis, the mutants fell into two different categories. The first category (6/7 variants) is composed of those substitutions that are likely to have a direct or indirect effect on protein structure, owing to a loss of a stabilizing interaction with one or more nearby residues. Substitutions falling into this category are p.Gln400Pro, p.Arg404Trp,

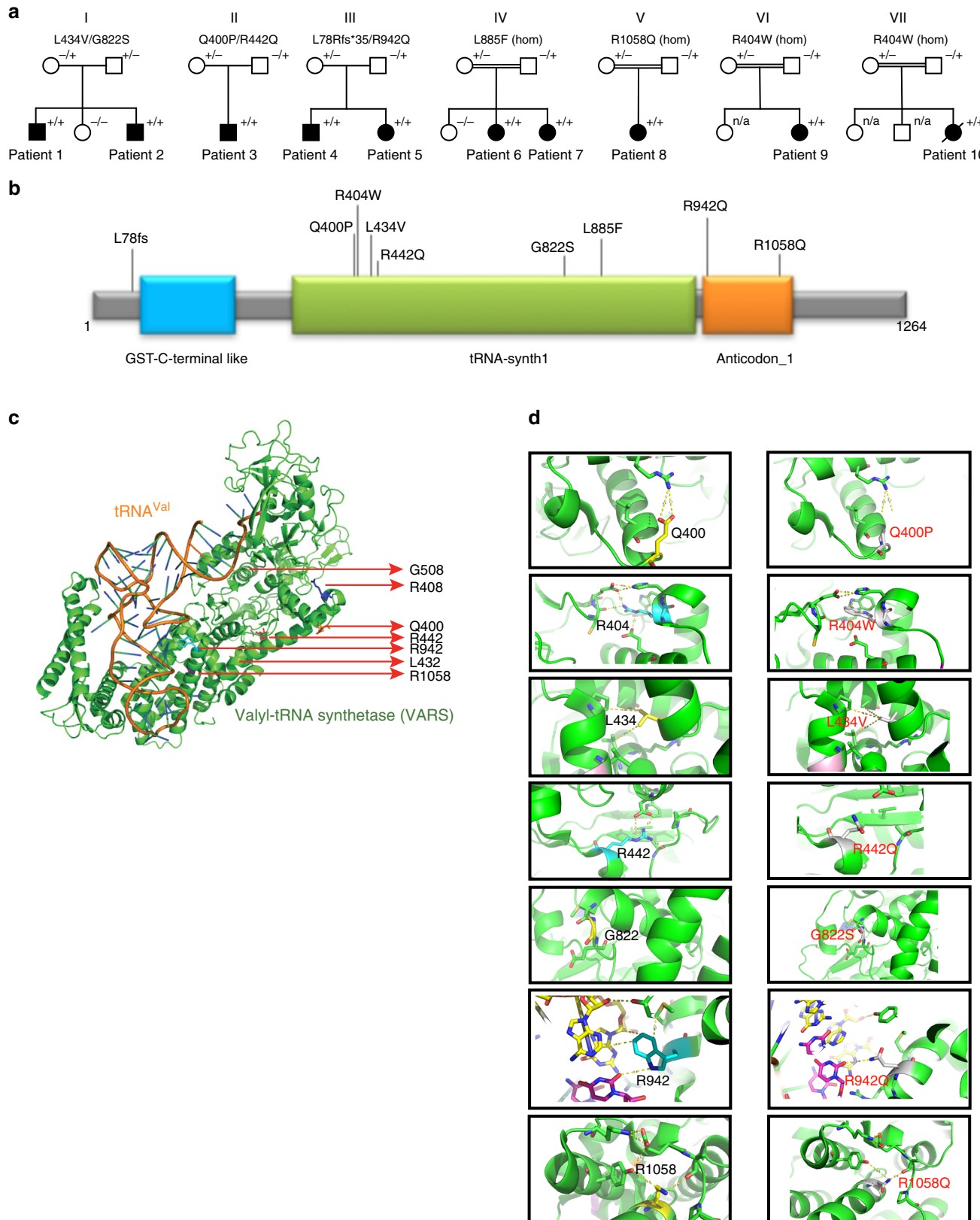

**Fig. 1** Identification of *VARS* variants in seven families with developmental encephalopathies and in silico predictions. **a** Pedigrees of the seven families diagnosed with *VARS* mutations. **b** Location of the identified *VARS* variants on protein level (InterPro/P26640). **c** Ribbon cartoon model of the *Thermus thermophilus* VARS-tRNA complex, highlighting the residues corresponding to those substituted in the human model. **d** Pair-wise comparisons between the wild-type (left) and mutant (right) residues for predicted changes in local contacts with tRNA or other amino acids. Hydrogen bonds were indicated as dotted yellow lines

p.Leu434Val, p.Arg442Gln, p.Gly822Ser, and p.Arg1058Gln. Prediction was weakest for p.Gln400Pro and p.Arg1058Gln as the affected residues do not make particularly strong interactions with other nearby residues. The second category (1/7) is defined by the single mutant substitution (p.Arg942Gln), which alters a direct contact to the transfer RNA substrate and, as such, could interfere with tRNA binding. The remaining variant, p.Leu885-Phe, is located in an insertion in helix α13 that is missing in *T. thermophilus* ValRS and could not be modeled directly. Inspection of the structure of the complex suggests that this is located potentially close to the N4 of the tRNA anticodon nucleotide C38 in the bound complex and may affect one or more direct contacts to anticodon functional groups.

**In vitro modeling in yeast and patient-derived cell lines.** Fibroblast lines of the two siblings of family III (patients 4 and 5) with the compound heterozygous p.Leu78Argfs*35/p.Arg942Gln variants were available for functional testing. Both cell lines showed a statistically significant approximately twofold decrease in VARS protein compared to a control line (Fig. 2a, $p < 0.001$, one-way ANOVA). We hypothesized that this observation was consistent with nonsense mediated mRNA decay (NMD) of the mRNA bearing the frameshift allele and experimentally tested this hypothesis by performing RT-qPCR before and after the fibroblasts were treated with a translation inhibitor, cyclohex-imide (CHX)[29]. In the patient fibroblasts there was a clear absence of the frameshift allele at the cDNA level, which could be reversed by CHX treatment (Fig. 2b). Immunofluorescence staining further demonstrated that VARS is predominantly ER-associated and that this localization was retained in patients 4 and 5 (Fig. 2c and Supplementary Figure 2). Extracts from these cell lines were further assessed for their ability to support aminoacylation. VARS activity from these mutated cell lines was significantly reduced (<25%) relative to control fibroblasts, whereas TARS activity was not, demonstrating that the observed reduction in enzymatic activity was VARS specific (Fig. 2d).

To further assess the functional effects of pathogenicity of the identified *VARS* missense variants, yeast complementation assays were performed by modeling *VARS* variants in the *S. cerevisiae* ortholog, *VAS1*. Complementation assays were performed for all variants that were conserved between human and yeast (p.Leu434Val, p.Arg442Gln, p.Gly822Ser, and p.Leu885Phe) on a haploid yeast strain with the endogenous *VAS1* gene deleted. Viability of this strain was maintained via a *URA3*-bearing vector (pRS316) with a wild-type copy of *VAS1*. Wild-type *VAS1* on pRS315 supported yeast growth while the empty pRS315 vector did not (Fig. 2e) consistent with our experimental vector harboring a functional *VAS1* allele and with *VAS1* being an essential gene. The assays further showed that p.Leu434Val, p. Arg442Gln, and p.Leu885Phe *VAS1* alleles sustained yeast growth to a level comparable to wild-type *VAS1* (Fig. 2e). In contrast, p. Gly822Ser did not support yeast cell growth at all (Fig. 2e) consistent with p.Gly822Ser *VARS* being a functionally null allele. These mutants were further tested for their ability to support VARS enzymatic activity. Patient-derived lymphoblast lines were available for patients 1 and 2 carrying the p.Leu434Val/p. Gly822Ser variants, their heterozygous parents, and patient 9, who was homozygous for the p.Arg404Trp variant. As shown in Fig. 2f, VARS aminoacylation was decreased on the order of 50% in patients 1 and 2 relative to that of their parents ($p < 0.05$, one-way ANOVA), whereas TARS aminoacylation was >100% of the parental values. The p.Arg404Trp homozygote (patient 9) variant showed an even greater loss of VARS activity, on order of 25% relative to the heterogeneous parental samples from family I.

**Development of a *vars* knockout zebrafish model.** The zebrafish genome encodes a single *VARS* orthologue (67% identity and 77% similarity at the amino acid level to human) with a highly conserved catalytic domain (80% identity and 89% similarity). In order to deduce the potential role of *vars* during development, its expression patterns were examined during early development (Fig. 3a). *vars* mRNA was found to be ubiquitously expressed at 18-somite stage at 18 hours post fertilization (hpf), with more distinctive expression in the brain region and in the prospective eye as well as in the hematopoietic intermediate cell mass and somites, which was maintained till 24 hpf. From 36 hpf the expression of *vars* became restricted to the developing brain, and after 48 hpf it was also observed in other developing organs, including branchial arches, liver, pancreas, and intestine (Fig. 3a and Supplementary Figure 3). These dynamic expression patterns strongly suggest an essential role of *vars* in the brain development, while the expression outside CNS also suggests multiple roles of *vars* during organogenesis.

To examine potential functional effects of *VARS* LoF in vivo, we generated a *vars* knockout model using CRISPR/Cas9 technology in zebrafish[30,31] (Supplementary Figure 4). Via RT-qPCR, we confirmed that there was an almost complete lack of total *vars* mRNA in *vars*−/− larvae at 3 and 5 days post fertilization (dpf) and approximately half of the transcript present in *vars*+/− larvae, proving the efficiency of the *vars* knockout (Fig. 3b). Absence of *vars* in *vars*−/− larvae led to premature death between 8 and 12 dpf (*vars*+/+ and *vars*+/− siblings remained alive), suggesting an essential role for VARS in survival (Fig. 3c). At 7 dpf about 35.3% of *vars*−/− larvae showed no touch response, which was in marked contrast to the increasing touch response demonstrated by *vars*+/+ and *vars*+/− over the course of development (Fig. 3d). Loss of posture could be observed in about 88.2% of *vars*−/− larvae already at 6 dpf (Fig. 3d) and from 6 to 8 dpf ~50% of *vars*−/− larvae displayed abnormal motor behavior such as jerky spasmodic movements during touch response, in contrast to *vars*+/+ and *vars*+/− (data not shown). Severe morphological abnormalities were observed in *vars*−/− larvae from 3 dpf onwards, which were progressing over the course of time. The most prominent features were microcephaly with a partial loss of forebrain and snout, microphthalmia, and pericardial edema, as shown in bright-field images (Fig. 3e and Supplementary Figure 5a). These dysmorphologies were further investigated at the histological level on zebrafish forebrain sections from 1 to 5 dpf (Fig. 3f). In 2 dpf *vars*−/− larvae, aberrant cells desquamated into the ventricular space of the forebrain and midbrain, which indicates substantial cellular changes at an early stage of development (Supplementary Data 2) that was not observed in the *vars*+/+ and *vars*+/− siblings. Structural abnormalities progressed over time resulting at 5 dpf in disrupted brain architecture, reduced jaw structures, delayed retinal lamination, reduced lens, and periocular swelling around the eye (corneal edema) (Fig. 3f and Supplementary Data 2). Measurements taken from different head areas identified significant reductions ($p < 0.0001$, one-way ANOVA) in head and eye size at 3 dpf (Fig. 3g, i), that became even more prominent at 5 dpf, where also the brain size was smaller compared to *vars*+/+ and *vars*+/− siblings (Fig. 3g–i).

To determine whether the microcephaly was due to excessive cell death in the brain, we assessed apoptosis by using antibody to active caspase-3. There were significantly more apoptotic cells detected in 3–5 dpf *vars*−/− brains ($p < 0.001$, one-way ANOVA) than in those of their +/+ and +/− siblings (Supplementary Figure 5b), whereas no statistically relevant difference was observed at 2 dpf. Collectively, these results suggest that loss of *vars* expression compromises zebrafish head and eye development and that Vars is essential for neuronal survival.

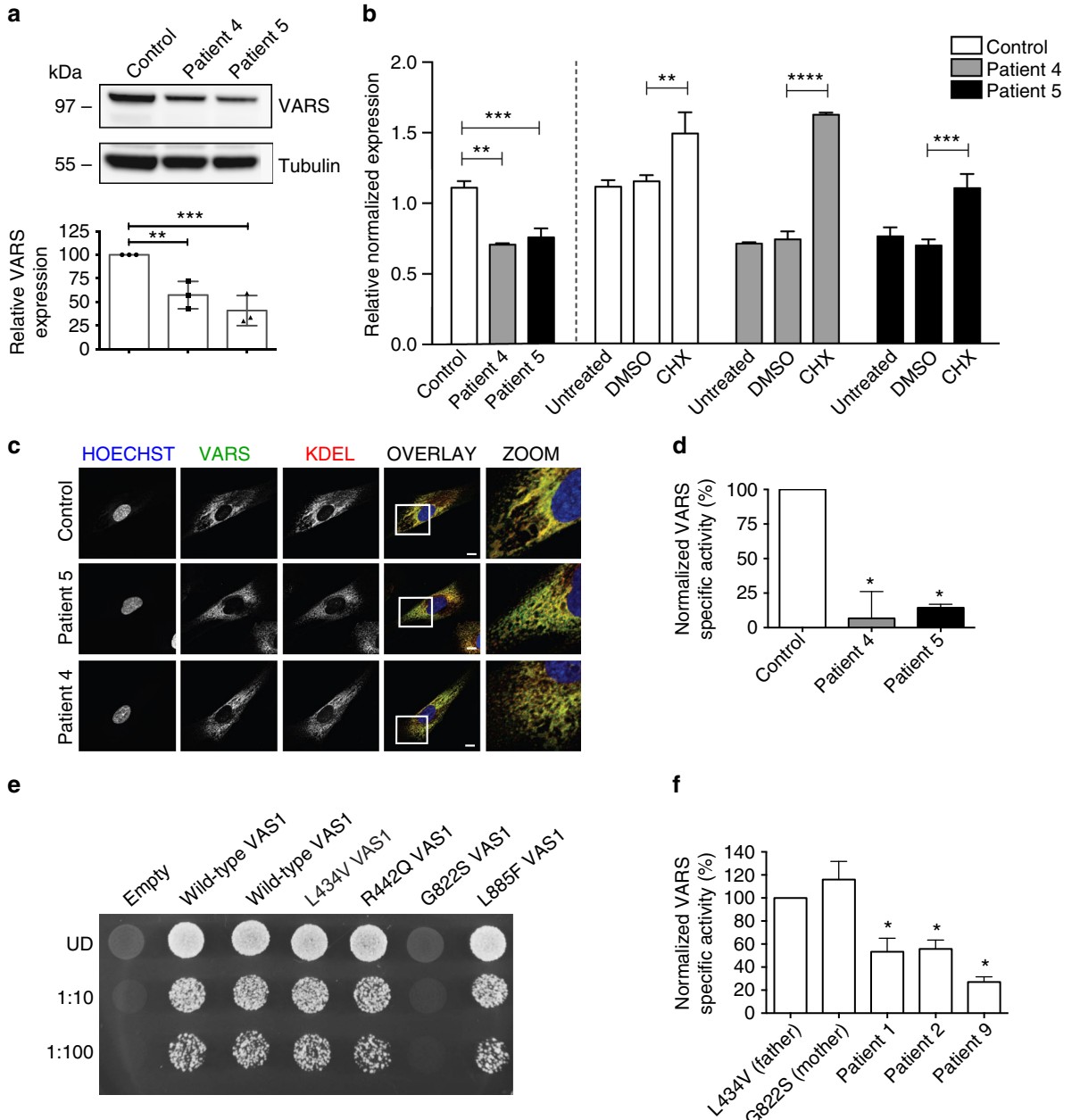

**Fig. 2** In vitro studies support variant pathogenicity. **a** Western blot performed on patient-derived fibroblasts from patients 4 and 5 of family 3 carrying the L78Rfs*35/R942Q *VARS* variant showed almost 50% reduction in VARS protein. Values are mean of three separate experiments. Error bars represent SD. **b** RT-qPCR on the fibroblasts showed almost complete absence of the frameshift allele at mRNA level, treatment with cycloheximide caused a partial increase in expression of the frameshift allele, which was not seen with DMSO-treated control. Values are mean of three separate experiments performed in triplicate. Error bars represent SD. **c** Immunocytochemistry highlighted the nucleus (Hoechst), VARS and KDEL, a marker for endoplasmatic reticulum. VARS co-localizes with KDEL. There was no difference in localization between the control line and the patient fibroblasts. **d** VARS and TARS aminoacylation activity measured in extracts from the patient fibroblasts. The data were normalized to ATTC fibroblasts. VARS aminoacylation activity was measured in technical triplicate at three separate passages, and TARS activity was measured once in technical triplicate. Data are represented as mean-specific activity and error bars represent SEM. * indicates significant difference from control. **e** A haploid yeast strain deleted for endogenous *VAS1* was transformed with a *LEU2*-bearing pRS315 vector containing wild-type *VAS1*, the indicated mutant form of *VAS1*, or no insert (empty). Cultures for each strain (labeled along the top) were either undiluted (UD) or diluted 1:10 or 1:100 and then spotted on solid medium containing 5-FOA to determine whether the *VAS1* alleles complement loss of endogenous *VAS1* at 30 °C. Only G822S shows absent growth indicating a functional null allele. **f** VARS and TARS aminoacylation activity measured in extracts from patient-derived lymphoblasts of patients 1 and 2 (L434V/G822S) and their parents and patient 9 (R404W). Data were normalized to the paternal cells. VARS aminoacylation activity was measured in technical triplicate at three separate passages, and TARS activity was measured once in technical triplicate. Data are represented as mean-specific activity and error bars represent SEM. * indicates significant difference from L434V paternal lymphoid cells. In **a**, **b**, **d**, and **f** one-way ANOVA with Tukey's multiple comparisons test was used. Significant values are noted **$p < 0.01$, ***$p < 0.001$, and ****$p < 0.0001$

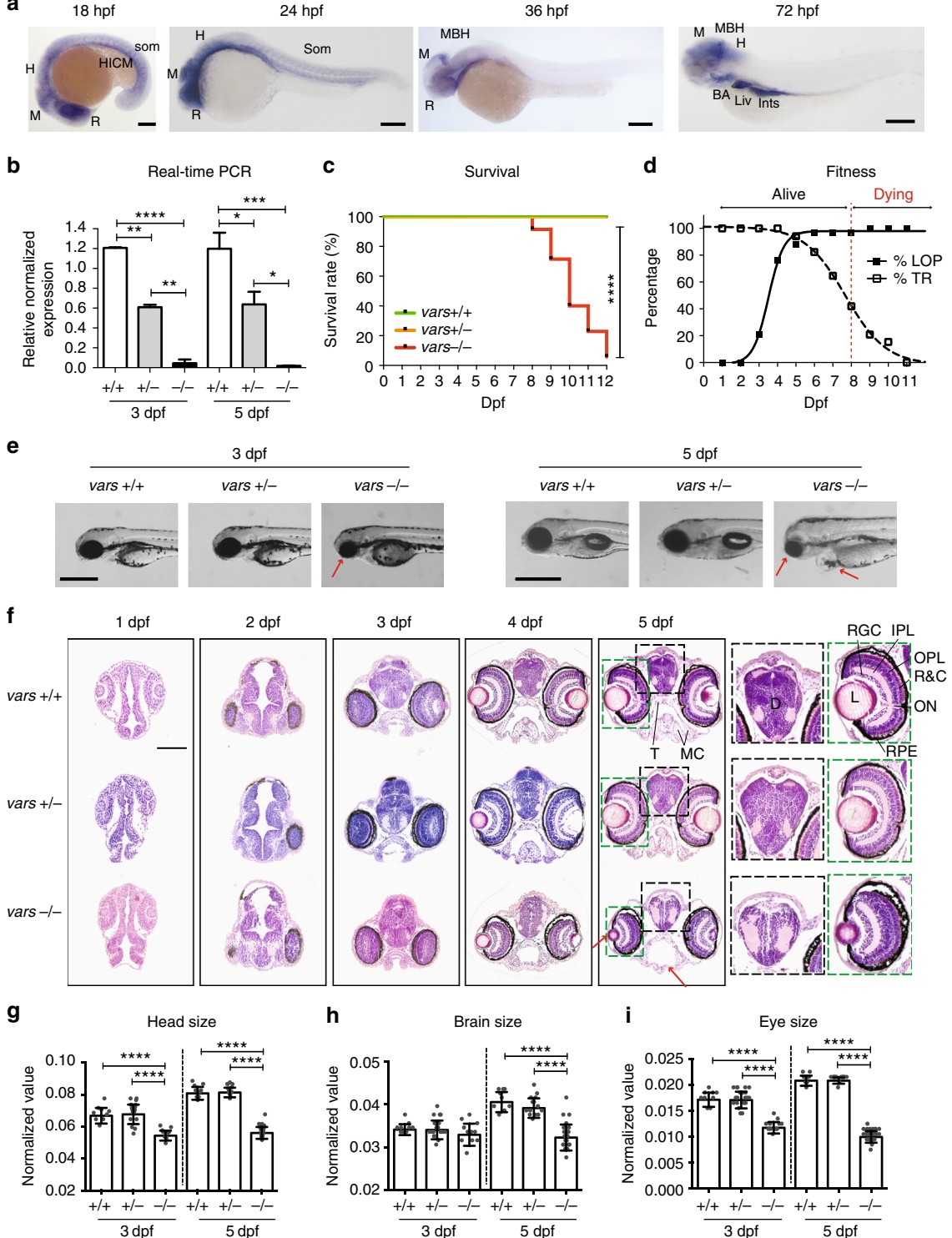

**Behavioral and electrophysiological studies in zebrafish.** Changes in behavioral activity were investigated by performing tracking experiments. *vars−/−* larvae showed significantly decreased swimming activity from 4 till 7 dpf ($p < 0.0001$, one-way ANOVA) in comparison to *vars+/+* and *vars+/−* (Fig. 4a). In addition, we examined if *vars−/−* larvae possessed learning deficits by performing a habituation assay[32]. In the experiment, 6 dpf larvae were subjected to multiple dark flashes (DFs) alternated with light periods. Similarly to *vars+/+* and *vars+/−*, *vars−/−* larvae could adapt to DFs during a training period as

shown by their movement values decreasing over time (Fig. 4b). Interestingly however, the motion of *vars* knockout larvae in response to DF was significantly increased ($p < 0.0001$, one-way ANOVA).

To investigate whether *vars* knockout resulted in abnormal brain activity electrographically, we recorded local field potentials (LFPs) on larval optic tecta of 5, 6, and 7 dpf *vars−/−*, *vars+/−*, and *vars+/+* zebrafish larvae. Epileptiform events were defined as multi-spike bursts with amplitudes equal to or exceeding threefold the baseline (Fig. 4c). Recurrent spontaneous

**Fig. 3** vars−/− larvae display severe developmental phenotype with early lethality. **a** Spatiotemporal expression patterns of vars by whole-mount RNA in situ hybridization at 18, 24, 36, and 72 hpf. BA Branchial Arches; H Hindbrain; HICM Hematopoietic Intermediate Cell Mass; Ints Intestine; Liv Liver; MBH Midbrain-Hindbrain boundary; M Midbrain; R Retina. Scale bars = 200 μm **b** RT-qPCR data demonstrating the expression of total vars in vars+/+, vars+/−, and vars−/− larvae at 3 and 5 dpf. Values are mean of three separate experiments performed in triplicate. **c** Kaplan–Meier survival curve of vars+/+ (n = 37), vars+/− (n = 82), and vars−/− (n = 34) larvae. **d** Graph illustrating changes in loss-of-posture and touch response of vars−/− larvae throughout the life span. Only the surviving larvae were included. **e** Representative lateral and dorsal bright-field images of 3 and 5 dpf vars+/+, vars+/−, and vars−/− larvae (scale bar, 500 μm). Pericardial edema, small eye, and periocular swelling around the eye were marked with arrows. **f** H&E histological staining of paraffin-embedded coronary sections from the forebrain of 1–5 dpf vars+/+, vars+/−, and vars−/− larvae (scale bar, 100 μm). Magnification of the disruption in the organization of the brain and the eye for 5 dpf vars+/+, vars+/−, and vars−/− was marked with black and green stripped line, respectively. Red arrows point out some structural abnormalities. D diencephalon; IPL inner plexiform layer; L lens; MC mandibular cartilage; ON optic nerve; OPL outer plexiform layer; R&C rods and cones; RGC retinal ganglion cell; T trabecula. **g–i** Comparison of the individual measurements for head size **g**, brain size **h**, and eye size **i** for vars+/+, vars+/−, and vars−/− at 3 dpf (n = 9, n = 20 and n = 11, respectively) and 5 dpf (n = 9, n = 14 and n = 24, respectively). In **b** and **g–i** one-way ANOVA with Tukey's multiple comparisons test was used. Values are mean of three separate experiments. In **c** log-rank (Mantel-Cox) test was used. Error bars represent SD. Significant values are noted as ***$p < 0.001$ and ****$p < 0.0001$

epileptiform events occurred in vars−/− at all days, being most prominent at 5 dpf where 68.57% of the larvae displayed abnormal activity, whereas no vars+/− and only 5.26% of vars+/+ showed similar events (Fig. 4d). At 6 and 7 dpf, in 47.62 and 52.63% of vars−/− larvae, respectively, electrographic seizure activity could be detected.

The morphological and behavioral phenotype of our vars−/− CRISPR model was fully recapitulated and confirmed in the vars^Hi558Tg−/− zebrafish line, bearing a retroviral insertion in the first intron of vars (Supplementary Figure 6).

**Rescue experiments with human VARS mRNA.** To further validate the specificity of the vars knockout phenotype traits with respect to modeling neurological disease and to determine the functional consequence of selected variants identified in patients, we performed rescue experiments through mRNA injections of either wild type or mutated human VARS into vars−/− CRISPR embryos. Supplying WT VARS led to a partial or full rescue of the early phenotype in vars−/− larvae. In early development (3 dpf), a statistically significant increase for the head, brain, and eye size was observed ($p < 0.001$, one-way ANOVA, Fig. 5a–c, respectively). A significantly ameliorated touch response ($p < 0.001$, one-way ANOVA, Fig. 5d), occurred in tandem with the improvement of locomotor activity being most prominent at 5 dpf (Fig. 5e), suggesting a late response. These results were supported by the presence of exogenously supplied human WT VARS mRNA detected in vars−/− larvae at 1, 3, and 5 dpf (Fig. 5f).

In order to provide further evidence for the disease-causing nature of some of the variants, human VARS mRNA with mutant substitutions in the catalytic (p.Gln400Pro) and anticodon binding domains (p.Arg942Gln and p.Arg1058Gln) were tested for their ability to complement the vars knockout phenotype. For this experiment, we employed the early phenotype readout at 3 dpf. In contrast to the results with WT human VARS mRNA, none of the injections of human mutated VARS mRNA rescued head size or brain size of vars−/− larvae (Fig. 5g, h, respectively). Among the three variants, only the p.Arg942Gln variant provided a partial rescue of the eye size phenotype (Fig. 5i). These observations confirmed that, despite differences in their predicted impact on VARS structure and mechanism, all three variants displayed a loss of function when tested in our vars zebrafish model.

## Discussion

Aminoacyl-tRNA synthetases have an indispensable function in protein translation, which explains their extensive conservation throughout evolution and the fact that mutations in this class of proteins have been increasingly associated with human disease. To date, a total of 31 out of 37 ARS enzymes have been implicated

in genetic diseases, which often include neurological features[2]. We studied ten patients with biallelic VARS variants and report seven new variants. Detailed clinical evaluation revealed developmental delay, early onset epilepsy, and microcephaly as key clinical characteristics. While no specific epilepsy syndrome could be associated with VARS mutations, a single patient (patient 2) had an EEG confirmed diagnosis of epilepsy of infancy with migrating focal seizures (EIMFS). Notably, this clinical diagnosis was also found in a patient with a recessive QARS mutation[18], suggesting a potential link between this rare type of epilepsy and dysfunction of aminoacyl-tRNA synthetases. Two patients (patients 4 and 5) never manifested seizures. Interestingly, these two patients had a normal brain MRI, whereas cerebral atrophy, hypoplasia or atrophy of the corpus callosum and hypomyelination or delayed myelination were recurrent findings in the other patients. Both siblings also had severe intellectual disability to the same degree as most other VARS patients. Clinical features closely resembling the phenotype to the patients with VARS mutations described in this study have been observed for mutations in other members of the cytoplasmic ARS family, most striking in patients with biallelic mutations in QARS[18,19], KARS[17], and AARS[20,33]. This may indicate a joint phenotypic spectrum associated with certain aminoacyl-tRNA synthetases defects that predominantly involves developmental delay, epilepsy, and microcephaly. Interestingly, also the phenotype associated with biallelic mitochondrial VARS2 variants showed some similar features[34]. This illustrates the importance of including a broad set of genes involved in protein translation in the screening of patients with developmental (and epileptic) encephalopathies.

Eight of the nine identified VARS variants were missense variants located in or near the catalytic or tRNA binding domains. The p.Leu78Argfs*35 variant, occurring in the compound heterozygous state with the p.Arg942Gln missense variant, was the only truncating VARS variant reported in our series. The frameshift allele was shown to be prone to NMD resulting in a clear decrease in expression of the mutated mRNA and protein, and decrease in VARS aminoacylation activity, consistent with a LoF allele. Immunofluorescence staining further showed that VARS is predominantly localized at the ER. This expression pattern was not altered in fibroblasts in the presence of the p.Leu78Argfs*35/p.Arg942Gln variants.

To gain further insight into the pathogenic nature of the missense variants we performed predictive in silico studies of all mutations. Next, we supplemented this preliminary mutational analysis with biochemical, developmental, and neurological trait studies of selected pathogenic variant alleles in yeast, patient-derived cell lines and/or zebrafish. While a full functional characterization of each missense mutant was not performed, combined predictive and focused biochemical studies of different

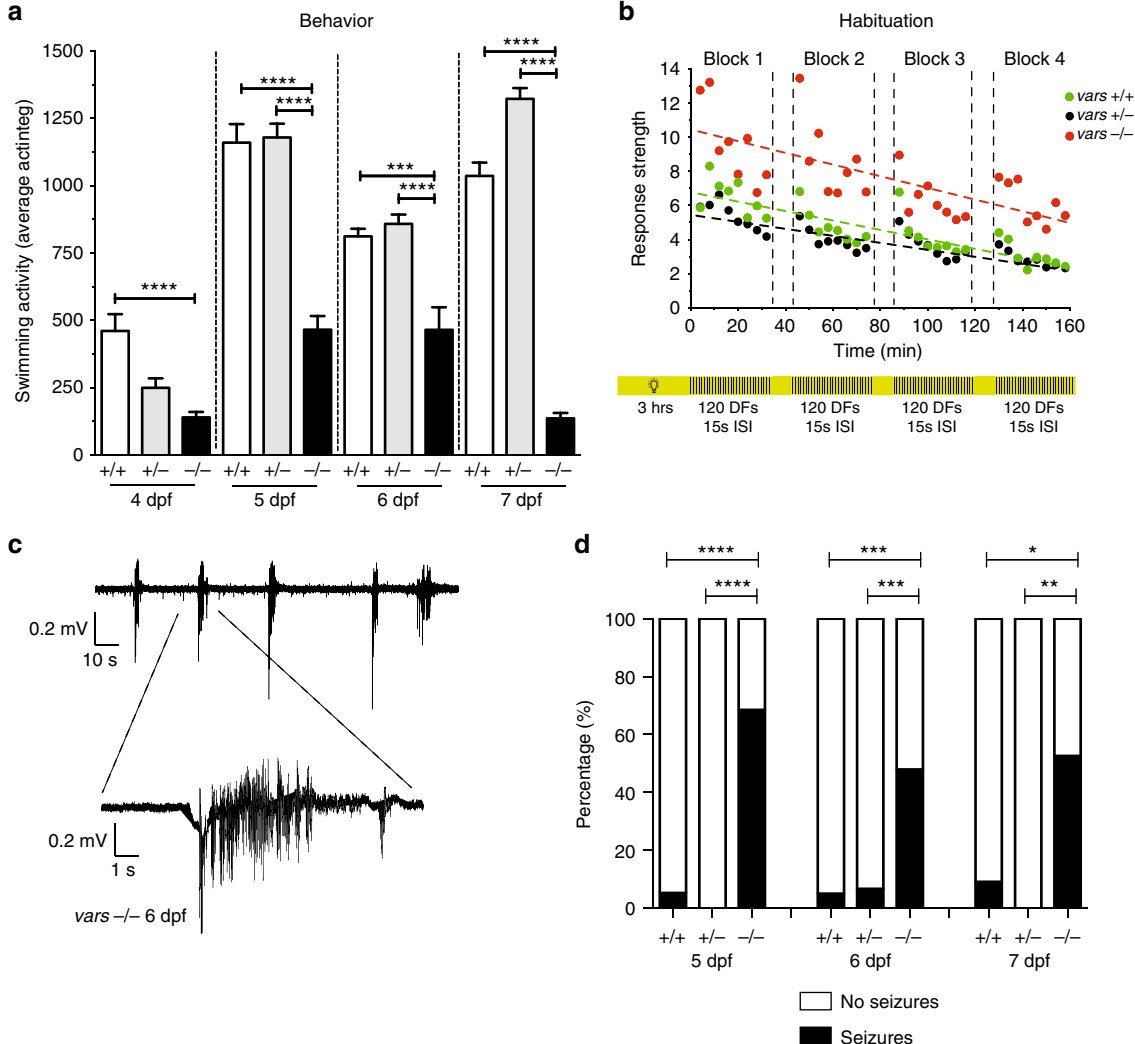

**Fig. 4** *vars*−/− larvae show cognitive deficits and spontaneous seizure like behavior. **a** Behavioral activity (average total movement) of *vars*+/+ (*n* = 58), *vars*+/− (*n* = 108), and *vars*−/− (*n* = 58) larvae from 4 to 7 dpf expressed in actinteg units. Values are mean of three separate experiments. **b** Habituation assay performed on 6 dpf *vars*+/+ (*n* = 19), *vars*+/− (*n* = 47), and *vars*−/− (*n* = 30) larvae was composed of four blocks with 120 DFs with 15 s ISI (regions with black and yellow stripes), alternated by 10 min of light (yellow areas). The linear slopes between the actinteg responses were calculated (dotted lines) and compared between the different genotype groups (green circles—*vars*+/+, black circles—*vars*+/−, and red circles—*vars*−/−). DF dark flashes, ISI interstimulus intervals. Values are mean of two separate experiments. **c** Representative recording from optic tectum of 6 dpf *vars*−/− larva displaying polyspike discharges. Top trace represents typical pattern of epileptiform activity. Bottom trace shows magnification of the epileptiform event. **d** Percentage of larvae exhibiting spontaneous electrographic activity recorded from 5 dpf *vars*+/+ (*n* = 19), *vars*+/− (*n* = 31), and *vars*−/− (*n* = 35), 6 dpf *vars*+/+ (*n* = 20), *vars*+/− (*n* = 30) and *vars*−/− (*n* = 48) and 7 dpf *vars*+/+ (*n* = 12), *vars*+/− (*n* = 13) and *vars*−/− (*n* = 19) larvae. Abnormal brain activity was observed in 68.57% (24/35) 5 dpf *vars*−/−, 5.26% (1/19) 5 dpf *vars*+/+, 47.62% (23/48) 6 dpf *vars*−/−, 5% (1/20) 6 dpf *vars*+/+, 6.67% (2/30) *vars*+/− and 8.33% (1/12) 7 dpf *vars*+/+. In **a** and **b** one-way ANOVA with Tukey's multiple comparisons test was used. In **d**—Fisher's exact test was used. Significant values are noted *$p < 0.05$, **$p < 0.01$, ***$p < 0.001$, and ****$p < 0.0001$. Error bars represent SD

variants gave us sufficient information to make a statement about the pathogenic nature of most variants. Substitutions p.Gly822-Ser, p.Leu434Val, p.Arg442Gln and p.Arg404Trp were each characterized by a potential destabilization of structure. While stability was not investigated directly, the results of yeast complementation analysis indicated that p.Gly822Ser clearly failed to complement, while p.Leu434Val and p.Arg442Gln were still positive for complementation. Aminoacylation assays on patient-derived cell lines further showed a clear decrease in function for p.Leu434Val, p.Gly822Ser and p.Arg404Trp. On the other hand, substitutions p.Arg942Gln and p.Leu885Phe were predicted to cause impaired recognition of substrates. The predicted tRNA binding mutant p.Arg942Gln exhibited only weak

rescue of VARS function in the zebrafish knockout, and also contributed to the decreased aminoacylation seen in cell lines derived from the siblings carrying the variant. As such, this variant likely represents a hypomorphic *VARS* allele. The p.Leu885Phe variant was positive for yeast complementation, but was not tested for its ability to rescue the *vars*−/− zebrafish. Modeling analysis for the two missense variants p.Gln400Pro, and p.Arg1058Gln predicted a slightly lower likelihood of a potential protein destabilization. Nevertheless, neither p.Gln400Pro nor p.Arg1058Gln were able to rescue the zebrafish *vars* knockout providing experimental evidence for variant allele pathogenicity. This observation highlights the value of the zebrafish in vivo model to differentiate between pathogenic and

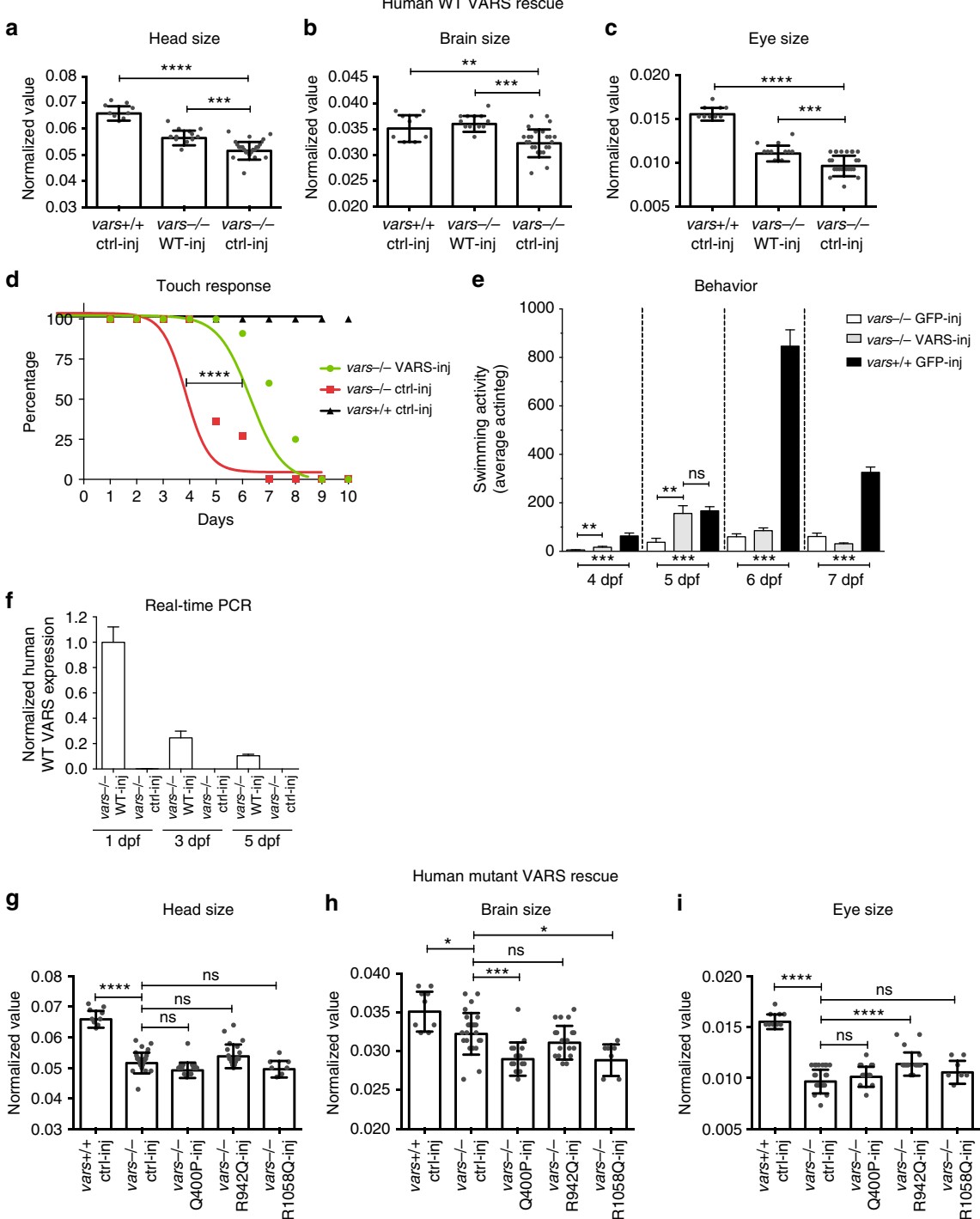

**Fig. 5** Wild-type human *VARS* mRNA partially rescues early zebrafish phenotype, whereas mutated constructs do not. **a–c** Comparison of the individual measurements for head size **a**, brain size **b**, and eye size **c** for WT human VARS-injected *vars*−/− larvae (*n* = 12). GFP-injected *vars*+/+ larvae (*n* = 9) were used as a positive control, whereas GFP-injected *vars*−/− larvae (*n* = 24) served as a negative control. Values are mean of three separate experiments. **d** Curves illustrating the evolution of touch response during the life span of *vars*−/− larvae injected with WT human *VARS* (*n* = 12) and GFP mRNA (*n* = 11) (negative control). *vars*−/− GFP-injected larvae were used as a positive control. **e** Average total movement of WT human *VARS*-injected (*n* = 10) and GFP-injected (*n* = 9) *vars*−/− larvae from 3 to 7 dpf. *vars*+/+ GFP-injected larvae (*n* = 14) were used as a control. **f** RT-qPCR data demonstrating expression levels of injected human WT *VARS* in *vars*−/− larvae at 1, 3, and 5 dpf. At 3 and 5 dpf there was 24.55% and 10.42% WT *VARS* mRNA left, respectively, in comparison to 1 dpf. Values are mean of three separate experiments performed in triplicate. **g–i** Comparison of the individual measurements for head size **g**, brain size **h**, and eye size **i** for human mutated Q400P (*n* = 15), R942Q (*n* = 18) and R1058Q (*n* = 7) VARS-injected *vars*−/− larvae. GFP-injected WT larvae (*n* = 9) were used as a positive control, whereas GFP-injected *vars*−/− larvae (*n* = 24) served as a negative control. Values are mean of three separate experiments. In **a–c** and **g–i** one-way ANOVA with Tukey's multiple comparisons test was used; in **d**—log-rank (Mantel-Cox) test; in **e**—unpaired *t*-test. Significant values are noted *$p < 0.05$, **$p < 0.01$, ***$p < 0.001$, and ****$p < 0.0001$. Error bars represent SD

potentially benign *VARS* variants with respect to the human neurological disease traits studied.

Based on the above observations, we hypothesize that biallelic disease-causing *VARS* variants consist of a combination of a loss-of-function allele and a hypomorphic allele or alternatively a combination of two hypomorphic alleles that decrease overall VARS levels below a threshold where protein synthesis and/or other VARS-specific functions are critically threatened. Homozygosity or compound heterozygosity for null alleles would be lethal due to the essential function of VARS in all cells. Importantly, this hypothesis is supported by our observation that certain disease-associated *VARS* alleles retain some function. For example, p.Leu434Val *VARS*, when modeled in yeast, allowed cellular growth, which is expected as this mutation is opposite the functional null allele p.Gly822Ser *VARS*. Similarly, p.Leu885Phe *VARS* allowed yeast cell growth, which is expected since this allele was identified in the homozygous state. It is important to point out that our yeast assay may not be able to detect more subtle defects in enzyme function as we have shown for *KARS* and *AARS* mutations associated with recessive disease[20,21].

While we acknowledge the *vars* knockout zebrafish model is not a perfect genetic model for the human disease, we showed it accurately recapitulated many of the phenotypic features. In particular, microcephaly and epileptic activity, as seen in the patients, could also be observed in the zebrafish. Moreover, hyperactivity in *vars*−/− larvae observed upon light switches suggests a neuromodulatory effect, as the behavior depends on sensory input and is regulated by neuronal connectivity of the hindbrain and spinal cord and multiple neurotransmitter pathways[35]. We showed that *vars* was strongly and selectively expressed in the developing brain, eye and other organs during embryogenesis, as such, it is not surprising that loss of *vars* expression severely compromised zebrafish development. Interestingly, a *qars* knockout and *iars* knockdown zebrafish model showed some features similar to the *vars* model including brain malformations and extensive cell death in the brain[18,36]. However, the behavioral and epileptic phenotype of the *qars* and *iars* model were not further investigated and therefore cannot be compared to our model. Noteworthy, microcephaly and microphthalmia seems to be a more general phenotype associated with ARS mutations in zebrafish, since several mutant ARS lines (*kars*, *mars*, *sars*, and *qars*), identified in a large insertional mutagenesis screen, displayed it[37].

Our results suggest that the pathomechanism of biallelic *VARS* variants, and possibly of recessive ARS disease more general, is a loss-of-protein function. This contrasts with dominant mutations in *ARS* genes which are nearly uniformly associated with Charcot-Marie-Tooth disease, or less common peripheral neuropathies, and their functional effects likely differ (hypothesized dominant negative or gain-of-function effect)[2]. This interpretation is further supported by the fact that parents of patients with recessive VARS-associated phenotypes, who carry the mutation in a heterozygous state, are consistently reported to be phenotypically normal. Indeed, a 50% reduction in VARS activity can be tolerated without significant pathology, which is also supported by the presence of heterozygous truncating variants in public databases like ExAC and gnomAD[26].

Noteworthy, pathogenic variants in other genes that encode proteins with a prominent role in translation, can cause similar neurological disease traits to the ones associated with pathogenic variants in ARSs. For instance, *CLP1*, which encodes an RNA kinase responsible for tRNA splicing required for tRNA maturation, causes microcephaly in humans, mice and zebrafish and peripheral neuropathy in humans[38,39].

Further studies on VARS, both in vitro and in the zebrafish model, may focus on gaining more functional insight into the underlying pathomechanism for the neurological disease traits focusing on the disrupted translational machinery and altered proteostasis or other non-canonical functions, and their effect on brain, nervous system, and global development. The zebrafish model further provides an excellent system for testing multilocus variation genetic models[40] and future compound screenings in the search for precision medicine directed therapies for these severe disorders.

## Methods

**Ethics.** All human research complied with all relevant ethical regulations and was approved by the relevant institutional review boards and the Ethical Committee of the University of Antwerp. Informed consents were obtained from all patients or their legal guardians in the respective centers where patients were recruited. Zebrafish experiments complied with all relevant ethical regulations and were approved by the Ethics Committee of the University of Leuven (Ethische Commissie van de KU Leuven, approval number ECD P150/2015) and by the Belgian Federal Department of Public Health, Food Safety and Environment (Federale Overheidsdienst Volksgezondheid, Veiligheid en de Voedselketen en Leefmilieu, approval number LA1210199).

**Genetic and phenotypic analysis.** Molecular genetic analyses were performed in different research and diagnostic centers for which the details are given in Supplementary Notes. Families 3 and 6 were ascertained through the program GeneMatcher[24]. The seven novel *VARS* variants were all identified with WES or WGS in patients for whom standard diagnostic work-up failed to identify the cause of their neurodevelopmental phenotype. Variant filtering was done through locally developed pipelines taking into account the quality of the variant calling, presence in population databases, predicted impact on the encoded protein and segregation of the variants. *VARS* transcript NM_006295.2 was used for variant nomenclature. Clinical information was collected using standardized clinical templates that were completed by collaborating clinicians and geneticists. International League Against Epilepsy (ILAE) criteria were used for epilepsy syndrome classification when applicable[41].

**In silico modeling of *VARS* variants.** The protein coding sequences of valyl-tRNA synthetases (VARS) were extracted from the UniProt server. The organisms and the UniProt accession numbers for their VARS sequences are as follows: *Escherichia coli* (E. coli), P07118; *Thermis thermophilus* (Thermis), P96142; *Saccharomyces cerevisiae* (yeast), P2637; *Drosophila melanogaster* (fruit fly), Q0E993; *Caenorhabditis elegans* (worm), Q23360; *Danio rerio* (zebrafish), F1Q740; *Mus musculus* (house mouse), Q9Z1Q9; *Homo sapiens* (human), P26640. Multispecies alignments were generated by use of Clustal Omega[42]. The alignment was further annotated in the protein ENDscript[43] with secondary structures derived from the structure of the ValRS-tRNA$^{Val}$ from *Thermis thermophilus* (PDB ID: 1IVS) as reported earlier[27,28]. The alignments were used to identify residues in the *Thermis thermophilus* complex corresponding to the residues altered by the VARS mutations, and then the resulting structures were rendered in space filling or stick representation using the program PyMOL (The PyMOL Molecular Graphics System, Version 1.2r3pre, Schrödinger, LLC).

**Isolation of patient-derived cell lines and cell culture.** Skin biopsies of patients 4 and 5 of family 3 (p.Leu78Argfs*35/p.Arg942Gln) and of a control individual were obtained following the local standard protocol and fibroblasts lines were generated. Fibroblasts, continuously kept at 37 °C in a humidified atmosphere with 5% $CO_2$, were cultivated in Dulbecco's Modified Eagle Medium (DMEM), high glucose (Gibco) enriched with 10% heat-inactivated fetal bovine serum (Gibco), 1% L-glutamine (Life Technologies), 1% penicillin-streptomycin (Life Technologies). Lymphoblast lines were generated from fresh (peripheral) blood samples for patient 1, 2 and their parents (p.Leu434Val/p.Gly822Ser); and patient 9 (p. Arg404Trp) according to local standard protocol. The cells, continuously kept at 37 °C in a humidified atmosphere with 6% $CO_2$, were cultivated in RPMI 1640 Medium (Gibco), enriched with 10% heat-inactivated fetal bovine serum (Gibco), 1% L-glutamine, 1% penicillin-streptomycin and 1% sodium pyruvate (Life Technologies).

**Western blot.** Pelleted fibroblasts of patients 4, 5, and a control individual were homogenized in lysis buffer (20 mM Tris-HCl, pH 7.4, 2.5 mM $MgCl_2$, 100 mM KCl, 0.5% NP-40) supplemented with protease inhibitors (Sigma), placed on ice for 30 min and cleared by centrifugation for 10 min at 20,800×g. Protein concentrations were determined using a Pierce BCA Protein Assay Kit (Thermo Fisher Scientific). Equal amounts of protein were diluted in NuPage LDS sample buffer 4X (Thermo Fisher Scientific) supplemented with 100 mM 1,4-dithiothreitol (DTT). Samples were denatured for 5 min at 95 °C. Subsequent size separation was performed with SDS-polyacrylamide gel electrophoresis on NuPAGE Novex 4–12% Bis-Tris gels (Thermo Fisher Scientific), which was later electrotransferred to a nitrocellulose membrane (GE Healthcare Lifescience). Membranes were blocked in

5% milk powder diluted in PBS-Tween 20 (0.1%) for 1 h. Primary antibody was incubated overnight at 4 °C or for 1 h at room temperature (RT) followed by secondary antibody for 1 h at RT. Visualization was effected with enhanced chemiluminescence detection using Amersham ECL Prime Western Blotting Detection Reagent (GE HealthCare) and an ImageQuantTM LAS4000 system (GE Healthcare Life Sciences). The density of the resulting bands, corrected for loading, was quantified using ImageJ and statistical significance assessed by one-way ANOVA with Bonferroni's multiple comparisons test. The following primary antibodies were used: anti-VARS antibody (1:2500, Atlas Antibodies, HPA046710) and anti-alpha-tubulin (1:5000, Abcam, ab4074). The following secondary antibodies were used: anti-rabbit HRP-conjugated (1:10,000, Promega, W401B), anti-mouse HRP-conjugated (1:10,000, Southern Biotech, 1070–05). Uncropped scans of blots are available in Supplementary Figure 7.

**qPCR of cycloheximide-treated cells**. Confluent T75 flasks of fibroblasts of patients 4, 5 and a control individual were treated for 6 h with (i) cycloheximide (CHX) (150 μg/mL), with (ii) dimethyl sulphoxide (DMSO) as a negative solvent control, or (iii) without treatment. After 6 h cells were collected, pelleted, and subsequently subjected to RNA extraction (Qiagen RNeasy Mini Kit), and followed by DNase treatment to remove residual genomic DNA (Turbo DNA free, Ambion). One microgram of total RNA was converted to cDNA using the Superscript® III First Strand Synthesis System (Life Technologies) with both oligo dT and random hexamer primers.

Primers for real-time detection of *VARS* were custom-made (Supplementary Table 3) and a reaction mixture containing 20 ng cDNA template, primers and Power SYBR Green PCR Mastermix (Life Technologies) was amplified under cycling conditions according to the manufacturer's protocol. Data were generated on a ViiATM7 Real-Time PCR system (Life Technologies) and analysed using Qbase+ (Biogazelle)[44]. *VARS* transcripts were normalized against 4 housekeeping genes (*GAPDH, HPRT1, SDHA*, and *HMBS*). The ΔΔCq method was used to determine the relative levels of mRNA expression between experimental samples and controls. The results consist of data from at least two separate experiments where samples were run in triplicate. One-way ANOVA with Tukey's multiple comparisons test was used to determine statistical significance.

**Immunocytochemistry**. For immunofluorescence (IF) staining of intracellular VARS protein, 50,000 fibroblasts of patients 4, 5 and a control individual were seeded on 12 mm diameter coverslips and 24 h later fixed with 4% paraformaldehyde (PFA) for 20 min at RT. Fibroblasts were permeabilized with 0.5% Triton X-100 in phosphate-buffered saline (PBS) for 2 min, blocked with 0.5% bovine serum albumin and 0.2% goat serum for 1 h and incubated overnight at 4 °C with following primary antibodies: anti-VARS protein (1:500; Atlas Antibodies, HPA046710), anti-KDEL (1:100; Enzo Life Sciences, 10C3), anti-Golgin-97 (1:200, Life Technologies, A21270), and anti-TOMM20 (1:200, Abcam, ab56783). With intermediate PBS washing steps, the secondary antibodies goat anti-rabbit IgG (Alexa Fluor 488) and goat anti-mouse IgG (Alexa Fluor 594) (both 1:500; Life Technologies) were added for 1 h at RT. Nuclei were stained for 10 min with Hoechst 33342 (1:10,000, Life Technologies). Coverslips were mounted (Dako) and images were taken with a Zeiss LSM700 confocal microscope using a ×63/1.40 plan-apochromatic objective. Possible cross-talk of the fluorescence channels was excluded by using frame-by-frame scanning.

**Aminoacylation assay**. Protein extracts containing aminoacyl-tRNA synthetase activity were prepared from ATTC control cells or patient-derived cell lines as described above. After washing twice with cold Dulbecco's PBS, cells were lysed in 50 mM Tris-HCl, pH 7.5, 150 mM NaCl, 5 mM DTT, 0.5% Triton X-100, and protease inhibitor cocktail (Sigma). Protein concentration was measured by the standard Bradford assay. Aminoacylation assays were performed at 37 °C in 100 mM HEPES, pH 7.2, 30 mM KCl, 10 mM MgCl$_2$ with 107 μM total human placental tRNA, 2 mM ATP, 50 μM [$^{14}$C] valine (282.8 mCi/mmol), and were initiated by the addition of protein extract to a final concentration of 0.3 μg/μL of total protein. At three different time points over a 10-min interval, 5 μL aliquots were spotted onto 3MM Whatman filter papers presoaked with 5% TCA. The dried filters were washed three times with 5% TCA, once with 95% ethanol, then the radioactivity was quantitated by liquid scintillation. To calculate the specific activity of each sample, aminoacylation rates (pmol aminoacylated tRNA/min) were calculated from linear fits of the progress curve data, corrected for the total protein concentration.

**Yeast complementation assay**. VAS1 expression constructs for yeast complementation assays were generated using Gateway Cloning Technology (Invitrogen). The *VAS1* gene (including the endogenous promoter sequence) was amplified from purified *S. cerevisiae* genomic DNA with primers bearing Gateway sequences. The resulting PCR product was BP-cloned into the pDONR221 entry vector per the manufacturer's instructions. The BP reaction was used to transform *E. coli*, colonies were purified and subjected to Sanger sequencing to confirm sequence specificity and the absence of PCR-induced errors. Mutations assayed were generated using the QuickChange II XL Site-Directed Mutagenesis Kit (Stratagene) and variant-specific mutagenic primers (Supplementary Table 3). Mutagenesis

reactions were performed on *VAS1* pDONR221 constructs, which were subsequently transformed into *E. coli*. DNA from individual clones was purified and subjected to Sanger sequencing to confirm the presence of each mutation and to rule out PCR-induced errors. Two sequence-validated constructs each for wild-type *VAS1* or the indicated mutations modeled in *VAS1* (Fig. 2d) were LR-cloned into pRS315 according to the manufacturer's instructions. LR reactions were then used to transform *E. coli*, and colonies were purified and digested with BsrG1 (New England Biolabs) to confirm the presence of the respective *VAS1* insert.

Yeast complementation assays were carried out using a haploid *S. cerevisiae* strain with a deletion of the endogenous *VAS1* locus. Viability of this strain was maintained via a pRS316 vector (including a *URA3* gene) bearing a wild-type copy of *VAS1*[45]. The haploid Δ*VAS1* strain was transformed with wild-type *VAS1*, mutant *VAS1*, or empty pRS315 bearing no *VAS1* insert; pRS315 includes a *LEU2* gene. Transformed yeast cells were selected for the presence of both vectors by growth on media lacking uracil and leucine. Two yeast colonies per transformation were selected for analysis and grown to saturation in 2 mL CM glucose broth (minus leucine and minus uracil) for 48 h at 30 °C. Cultures were either undiluted or diluted 1:10 or 1:100 in water. Subsequently, 10 μL of undiluted and diluted cultures were spotted on complete solid media containing 0.1% 5-fluoroorotic acid (5-FOA; Teknova) to select for spontaneous loss of the maintenance vector. Yeast viability was visually assessed after 3 days of incubation at 30 °C. Two colonies per transformation were assayed and each transformation was repeated three times using two independently generated constructs of either wild-type or the respective mutant *VAS1*.

**Zebrafish husbandry**. All zebrafish (*Danio rerio*) lines used in this study were maintained at 28.5 °C on a 14 h light/10 h dark cycle under standard aquaculture conditions in a UV-sterilized rack recirculating system equipped with a mechanical and biological filtration unit. Fertilized eggs were collected via natural spawning and were raised in Danieau's medium (1.5 mM HEPES, pH 7.2, 17.4 mM NaCl, 0.21 mM KCl, 0.12 mM MgSO$_4$, 0.18 mM Ca(NO$_3$)$_2$ and 0.6 μM methylene blue) in an incubator on a 14 h light/10 h dark cycle at 28.5 °C.

**Whole-mount RNA in situ hybridization (WISH)**. One kb coding sequence fragment of *vars* was amplified from cDNA of AB wild-type strain (primers in Supplementary Table 3) and cloned into Zero Blunt® TOPO® PCR Cloning Kit (Invitrogen). Cloned DNA was linearized by XhoI and HindIII, then synthesized by SP6 RNA polymerase and T7 RNA polymerase using DIG RNA labeling kit (all Roche) for sense- and anti-sense DIG-labeled RNA probes, respectively.

Embryos were fixed with 4% PFA, then washed with 1X PBS with Tween 20 (PBST), sequentially washed with 100–25% methanol and stored in 100% methanol at −20 °C until needed. On the first day of WISH, embryos were washed with 50–25% methanol, followed by 1X PBST. After treatment with proteinase K (Sigma) according to the developing stages, for permeabilization embryos were fixed again with 4% PFA and washed by 1X PBST. Embryos were hybridized in Hyb+ solution with the *vars* RNA probes at 70 °C overnight. On the second day, after serial washing with 2X SSCTw/50% formamide, 2X SSCT, 0.2X SSCT at 70 °C, embryos were blocked with 5% horse serum (Sigma) and incubated with anti-digoxigenin-AP Fab fragments (Roche) overnight at 4 °C. On the third day, embryos were developed with BCIP/NTP substrate (Roche). Staining was developed and stopped before the background signals started to appear in the embryos hybridized with the sense RNA probe.

**Generation of the *vars* CRISPR knockout zebrafish line**. A KO *vars* fish line was generated via CRISPR/ Cas9 technique[30,31]. *vars* sgRNA targeting exon 21 in the catalytic domain of vars protein (5′-CCGTCTCTAACAGTGTGCCC(GGG)-3′) was designed via GeneArt (Invitrogen) and further transcribed using MEGAshortscript™ T7 Transcription Kit (Ambion) and purified with MEGAclear™ Transcription Clean-Up Kit (Ambion). Cas9 (GeneArt CRISPR Nuclease mRNA) was purchased from Invitrogen. Single cell-stage fertilized wild-type embryos of AB line were injected with 7 pg *vars* sgRNA and 150 pg Cas9 mRNA (1 nL volume). The mutation at the target site was verified via Sanger sequencing. The remaining sgRNA/Cas9-injected embryos were raised till adulthood and outcrossed with WT adults. DNA extracted from F1 generation of 3 dpf whole larvae was screened for indels by Sanger sequencing. F0 founder with germline transmission and high rate of indels was selected to establish the knockout line. F1 generation embryos were raised to adulthood, fin clipped and sequenced. Individuals carrying the same mutation (4 bp deletion of GGGC) were identified and pooled together. All experiments were performed on embryos coming from F2 or F3 progeny.

To confirm the genotype of the larvae, prior to or at the conclusion of an experiment, a whole larva or fin clip, respectively, was placed in separate tubes with 50 μL of lysis buffer (100 μM Tris, 10 μM EDTA, 0.7 mM proteinase K and 0.2% Triton X-100) to extract genomic DNA. Lysis was performed at 55 °C for 3 h, followed by 10 min at 95 °C. Lysed samples were genotyped by performing a PCR to amplify a 466 bp-region of interest (containing *vars* 4 bp deletion) using Titanium® Taq DNA Polymerase (Takara) and *vars*-specific primers (Supplementary Table 3). Successfully amplified PCR products were purified using ExoProStar (Illustra™) and Sanger sequenced with the same primers used for amplification. The genotypes of the individual larvae were analysed using SeqMan software (LaserGene).

**mRNA extraction and RT-qPCR for zebrafish studies**. Total RNA from 1, 3, and 5 dpf *vars*+/+, *vars*+/−, and *vars*−/− larvae, respectively, was extracted using TRIzol (Ambion, Life Technologies). Residual genomic DNA was removed by treatment with DNase I (Roche) with Protector RNase inhibitor (Roche). Reverse transcription of total RNA to single-stranded cDNA was performed on 1 µg of total RNA using the High Capacity cDNA Reverse Transcription Kit (Applied Biosystems) and further diluted 1:20. Real-time PCR was performed in HardShell® Low-Profile Thin-Wall 96-Well Skirted PCR Plates (Bio-Rad) using CFX96 Touch Real-Time PCR Detection System (Bio-Rad). Primer and probe sequences for real-time detection of endogenous *vars*, as well as injected human WT *VARS* were custom-made (Supplementary Table 3). Reaction mixture containing diluted cDNA template, primers and 2x SsoAdvanced Universal SYBR Green Supermix (Bio-Rad) was amplified under cycling conditions according to the manufacturer's protocol. Data generated were analysed using CFX Manager Software (Bio-Rad). *vars* transcripts were normalized against ribosomal protein S18 (*rps18*) house-keeping genes that were experimentally determined to have the most stable expression in our reaction conditions. Primer and probe sequences for real-time detection of *vars* in Hi558Tg line were purchased from IDT (Supplementary Table 3). 2x TaqMan Universal Master Mix (ABI, USA) was used for the reaction. *vars* transcripts normalization was done against elongation factor 1-alpha (*ef-1 alpha*) and *rps18*. The ΔΔCq method was used to determine the relative levels of mRNA expression between experimental samples and controls. The results consist of data from at least two separate experiments run in triplicate. One-way ANOVA with Tukey's multiple comparisons test was used to determine statistical significance.

**Morphological studies in *vars*−/− zebrafish larvae**. Surviving *vars*+/+, *vars*+/−, and *vars*−/− larvae were counted daily and examined for major dysmorphologies, such as edema, head, and eye malformations, and for the presence of touch response, from 1 until 12 dpf. Dead larvae and all larvae from 12 dpf onwards were collected, stored at −80 °C and genotyped.

For head size measurements, *vars*+/+, *vars*+/−, and *vars*−/− larvae were positioned in 3% methylcellulose. Lateral images of the head and the whole body of 3 and 5 dpf larvae were acquired using Leica MZ 10F fluorescence microscope with a Leica DFC310 FX digital camera and Leica Application Suite V3.6 software. Measurements were done blinded, manually in ImageJ software. Body length was measured from the anterior tip of the snout to the base of the posterior caudal fin. Head, brain, and eye areas were measured by tracing the boundary of the surface of interest using some predetermined spots on the head such as the otic vesicle or dorsal indentation just above the eye, at the level of the pineal gland. The absolute values of measured surface were normalized to the total body length of the larvae. Results were analysed using one-way ANOVA followed by Tukey's multiple comparisons test.

**Histological analysis of *vars*−/− zebrafish larvae**. Fin clipped larvae were fixed in 4% PFA at 4 °C overnight and kept in 70% ethanol. At least five embryos or larvae per genotype group were embedded in 1% agarose in 1X TAE buffer. A mould, specifically designed to align zebrafish larvae, was used to produce agarose blocks with identical distributed wells of the same depth. Agarose blocks were gradually dehydrated in an enclosed automated tissue processor (Shandon Excelsior ES, Thermo Scientific) and subsequently embedded in paraffin. The heads of paraffin-embedded larvae were sectioned on a HM 325 manual rotary microtome (Thermo Fisher Scientific) at a thickness of 5 µm. The specimens were stained with hematoxylin and eosin (H&E stain) using Varistain™ Gemini ES Automated Slide Stainer (Thermo Fisher Scientific) according to laboratory protocols. The resulting sections were imaged at ×20 magnification in a SPOT 5.1 software (SPOT Imaging) by a SPOT-RT3 camera mounted on a Leica microscope. Brightness of the images was adjusted for the white background.

**Immunohistochemistry for active caspase-3**. IHC detection of cell death was carried out on 5-µm-thick deparaffinised and rehydratated sections. Prior to IHC, the specimens were subjected to heat-induced antigen retrieval by incubation in 10 mM sodium citrate (pH 6.0) for 10 min at 98 °C, followed by a 30 min cool down and treatment with 3% hydrogen peroxide (2 × 8 min). The sections were blocked for 30 min in 5% normal goat serum in 1X TBST, and further incubated with primary antibody against active caspase-3 (BD Biosciences, Clone C92-605, 1:500 dilution) for 1 h at RT. After rinsing with 1X TBST, HRP-conjugated secondary antibody (Jackson ImmunoResearch, 111-035-003, 1:200 dilution) was applied for 1 h at RT. Next, the slides were treated with DAB+/chromogen (DAKO) for 1 min at RT and rinsed with deionized water. Nuclear counterstain was performed in hematoxylin for 3 min. After clearing in ethanol and histoclear, the slides were coverslipped under mounting medium. For each staining a negative control was included by processing sections in the absence of the primary antibody. The images were taken at ×40 magnification in a SPOT 5.1 software (SPOT Imaging) by a SPOT-RT3 camera mounted on a Leica microscope. Three to four equivalent sections were selected for each group and DAB positively stained nuclei were counted using Fiji. The results were expressed as percentage of apoptotic cells of the total cell number within a selected brain area. Results were analysed using one-way ANOVA followed by Tukey's multiple comparisons test.

**Behavioral studies in *vars*−/− zebrafish larvae**. For the locomotor tracking, 3–7 dpf *vars*+/+, *vars*+/−, and *vars*−/− larvae were individually arrayed in a 96-well plate. After 30 min habituation, the larvae were placed in an automated tracking device (ZebraBox™, Viewpoint, Lyon, France) and their locomotor behavior was followed for 1 h under dark conditions. The total movement was quantified using ZebraLab software (Viewpoint, Lyon, France) and expressed in actinteg units. The results were analysed by one-way ANOVA with Tukey's multiple comparisons test.

For the habituation assay, 6 dpf *vars*+/+, *vars*+/−, and *vars*−/− larvae were individually arrayed in a 96-well plate and equilibrated for 3 h in a uniformly illuminated testing chamber in an automated video-tracking device (ZebraBox™, Viewpoint, Lyon, France). Further, they were exposed to a spaced training with light/dark cycles consisting of four blocks of 120 dark flashes (DFs) with 15 s interstimulus intervals (ISI) alternated with 10 min light periods. The increased movement during DFs was expressed in actinteg units. For the sake of clarity, standard deviations were not shown in the graph. Linear regression was used to find the best-fitting straight line through all the data points for each genotype group.

**Non-invasive local field potential recordings in *vars*−/− zebrafish larvae**. A larva was embedded in 2% low melting point agarose (Invitrogen). Recording electrodes were pulled from soda lime glass capillaries (1412227, Hilgenberg, Germany) on a DMZ Universal Puller (Zeitz, Germany) to a diameter of ~20 microns. It was filled with artificial cerebrospinal fluid (ACSF, 124 mM, NaCl, 2 mM KCl, 2 mM $MgSO_4$, 2 mM $CaCl_2$, 1.25 mM $KH_2PO_4$, 26 mM $NaHCO_3$, and 10 mM glucose) and placed on larva's head above the optic tectum. The recordings were performed using WinEDR (John Dempster, University of Strathclyde, UK). Differential signal was amplified 10,000 times by DAGAN 2400 amplifier (Minnesota, USA), band pass filtered at 0.3–300 Hz and digitized at 2 kHz via a PCI-6251 interface (National Instruments, UK).

All the larvae used for LFPs displayed touch response. The duration of each recording was 10 min. An electrical discharge was classified as a positive event when its amplitude was at least three times the amplitude of the baseline, and had a duration of at least 100 ms. The analysis of the epileptiform events was done using automated detection software that was previously developed and validated by our group[46].

**VARS rescue experiments**. Full-length wild-type human *VARS* cDNA (GenBank BC012808.2 from IRAUp969E0949D clone purchased from Source BioScience, UK) was cloned into a pCSDest vector (Addgene) using Gateway recombination technology according to the manufacturer's instructions (Life Technologies) (WT *VARS*-pCSDest). Single nucleotide variants, leading to substitutions p.Gln400Pro, p.Arg942Gln, and p.Arg1058Gln at protein level, were introduced into WT *VARS*-pCSDest vector via site-directed mutagenesis using PWO SuperYield DNA polymerase (Roche) and primers containing given variants (Supplementary Table 3). Subsequent DpnI digestion was done to remove parental plasmid. All templates were verified by direct Sanger sequencing.

Wild-type, p.Gln400Pro, p.Arg942Gln, and p.Arg1058Gln *VARS* mRNA was transcribed from linearized template plasmids using the SP6 mMessage mMachine® kit (Ambion) and purified by lithium chloride precipitation (Ambion). The rescue experiment was performed by cytoplasmic microinjection of 200 pg of WT or mutant *VARS* and GFP mRNA as control (1 nL volume) into 1-cell-stage *vars*−/− embryos. Following the injections, at 24 hpf only the healthy-looking embryos with normal morphology were selected for subsequent experiments (survival, behavioral tracking, and head measurements).

**Statistical analysis**. Data are presented as mean ± SD or mean ± SEM. Pairwise statistical significance was calculated with Student's unpaired *t*-test or Mann–Whitney test for data that failed the normality test and multiple comparisons were determined with one-way ANOVA with Tukey's test, using GraphPad Prism7 software.

## Data availability

All data generated or analysed during this study are included in this published article (and its supplementary information files). Human sequence (variant) data that support the findings of this study have been deposited in ClinVar and are accessible through the accession codes: ID:402133, ID:402134, SCV000808053, SCV000808054, SCV000808055, SCV000808056, SCV000808057, SCV000808058 and SCV000808059. All other relevant data are available from the corresponding authors on request.

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

## Acknowledgements

We would like to thank and acknowledge all patients and their families. We further want to thank Jolien Roovers and Pascal Simoens for their aid with sequencing of the zebrafish larvae and the cloning experiments; Michiel Krols, Sven Bervoets and Bob Asselbergh for their help with western blots and microscopy. We acknowledge the contribution of Peter De Rijk (as developer of GenomeComb—Antwerp pipeline) and the HPC facilities of the University of Luxembourg (http://hpc.uni.lu) for computational support and the VIB Genetic Support Facility for the zebrafish sequencing (https://www.molgen.ua.ac.be/Public/Organization/NSF/GSF.cfm). We also thank the Skeletal Biology and Engineering Research Center (KU Leuven, Belgium) (L.-A. Guns, A. Hens, L. Storms and E. Nefyodova) for help with histology experiments, Prof. Przemko Tylzanowski for assistance with cell death analysis and helpful discussions and Dr. Mario Wullimann and Prof. Alicia Ebert for their valuable comments concerning histological analysis. This study received funding from the Eurocores program EuroEPINOMICS, the Fund for Scientific Research Flanders (FWO) and the International Coordination Action (ICA) grant G0E8614N. It was further supported in part by the National Institute of Neurological Disorders and Stroke (R35 NS105078) to JRL and National Human Genome Research/National Heart Lung and Blood Institute (NHGRI/NHLBI) grant UM1 HG006542 to the Baylor Hopkins Center for Mendelian Genomics. The Center for Rare Childhood Disorders is supported through private donations made to the TGen Foundation. A.S. is a postdoctoral fellow of the Fund for Scientific Research Flanders (FWO, 12G3616N). H.S. is a PhD fellow of the Fund for Scientific Research Flanders (FWO, 1125416N). K.H. was funded by the Agency for Innovation by Science and Technology, IWT. J.-S.L. and H.K.G. were funded by Global Frontier project of the National Research

Foundation of Korea (NRF-M3A6A4-2010-0029785), National Research Council of Science & Technology (NST) grant of the Korea government (MSIP) (CRC-15-04-KIST), and KRIBB Research Initiative Program. F.D. was supported by a grant of the German Research Foundation/Deutsche Forschungsgemeinschaft (DI 1731/2-1). I.H. was supported by intramural funds of the University of Kiel, the German Research Foundation (HE5415/3-1) within the EuroEPINOMICS framework of the European Science Foundation and the German Research Foundation (DFG, HE5415/5-1, HE5415/6-1). C.F. is supported by a grant from the National Institute of General Medical Sciences (GM54899). A.A. is supported by a grant from the National Institute of General Medical Sciences (GM118647). The Solve-RD project has received funding from the European Union's Horizon 2020 research and innovation programme under grant agreement No 779257 (to V.T. and P.D.J.).

## Author contributions

P.D.J., P.d.W., and S.W. initiated the project which was subsequently developed and further coordinated also by A.S. and H.S. Patient recruitment, collection, and analysis of human DNA, patient-derived cell lines, and genetic data were carried out by H.S., T.D., K.H., D.N.S, F.D., K.L.H., J.H., S.I., E.J., E.K., O.E., R.S., S.R., J.R.L., R.K.N., P.M., V.N., M.P., K.R., S.V.S., I.H., and S.W. in collaboration with the C4RCD Research Group, and EuroEPINOMICS-RES AR working group. Patient fibroblasts were studied by H.S., T.D., and E.A. supervised by V.T. In silico modeling and aminoacylation assays were performed by C.F., P.C., and P.W. Yeast complementation assays were coordinated and performed by A.A. and S.N.O. Zebrafish experiments were carried out by A.S., M.P., J.S., Y.Z., A.K., J.-S.L., and H.K.G. supervised by P.d.W. Zebrafish genotyping was performed by H.S. and T.D. A.S., H.S., P.d.W., and P.D.J. wrote the manuscript with critical input from C.F. and A.A. All authors revised the final version of the manuscript.

## Additional information

Competing interests: J.R.L. has stock ownership in 23andMe, is a paid consultant for Regeneron Pharmaceuticals, has stock options in Lasergen, Inc., is on the Scientific Advisory Board of Baylor Genetic (BG) and is a co-inventor on multiple United States and European patents related to molecular diagnostics for inherited neuropathies, eye diseases, and bacterial genomic fingerprinting. The Department of Molecular and Human Genetics at Baylor College of Medicine derives revenue from molecular genetic testing offered in the BG Laboratories. All the remaining authors declare no competing interests.

Aleksandra Siekierska[1], Hannah Stamberger[2,3,4], Tine Deconinck[2,3], Stephanie N. Oprescu[5], Michèle Partoens[1], Yifan Zhang[1], Jo Sourbron[1], Elias Adriaenssens[3,6], Patrick Mullen[7], Patrick Wiencek[7], Katia Hardies[2,3], Jeong-Soo Lee[8,9,10], Hoi-Khoanh Giong[8,9,10], Felix Distelmaier[11], Orly Elpeleg[12], Katherine L. Helbig[13], Joseph Hersh[14], Sedat Isikay[15], Elizabeth Jordan[16], Ender Karaca[17,30], Angela Kecskes[1,31], James R. Lupski[17,18,19,20], Reka Kovacs-Nagy[21], Patrick May [22], Vinodh Narayanan [23], Manuela Pendziwiat[24], Keri Ramsey[23], Sampathkumar Rangasamy[23], Deepali N. Shinde[25], Ronen Spiegel[26,27], Vincent Timmerman[3,6], Sarah von Spiczak[24,28], Ingo Helbig[13,24], C4RCD Research Group, AR working group of the EuroEPINOMICS RES Consortium, Sarah Weckhuysen [2,3,4], Christopher Francklyn[7], Anthony Antonellis[5,29], Peter de Witte [1] & Peter De Jonghe[2,3,4]

[1]Laboratory for Molecular Biodiscovery, Department of Pharmaceutical and Pharmacological Sciences, KU Leuven, Leuven 3000, Belgium. [2]Neurogenetics Group, Center for Molecular Neurology, VIB, University of Antwerp, Antwerp 2610, Belgium. [3]Institute Born Bunge, University of Antwerp, Antwerp 2610, Belgium. [4]Department of Neurology, Antwerp University Hospital, Antwerp 2650, Belgium. [5]Department of Human Genetics, University of Michigan, Ann Arbor, MI 48109, USA. [6]Peripheral Neuropathy Research Group, Department of Biomedical Sciences, University of Antwerp, Antwerp 2610, Belgium. [7]Department of Biochemistry, University of Vermont, Burlington, VT 05405, USA. [8]Disease Target Structure Research Center, Korea Research Institute of Bioscience and Biotechnology, Daejeon 34141, Republic of Korea. [9]KRIBB School, University of Science and Technology, Daejeon 34141, Republic of Korea. [10]Dementia DTC R&D Convergence Program, Korea Institute of Science and Technology, Seoul 02792, Republic of Korea. [11]Department of General Pediatrics, Neonatology and Pediatric Cardiology, University Children's Hospital, Heinrich-Heine-University Düsseldorf, Düsseldorf 40225, Germany. [12]Monique and Jacques Roboh Department of Genetic Research, Hadassah-Hebrew University Medical Center, Jerusalem 01120, Israel. [13]Division of Neurology, Children's Hospital of Philadelphia, Philadelphia, PA 19104, USA. [14]Department of Pediatrics, Medicine, University of Louisville School of Medicine, 571S Floyd Street, Louisville, Kentucky 40202, USA. [15]Department of Physiotherapy and Rehabilitation, Hasan Kalyoncu University, School of Health Sciences, Gaziantep 27410, Turkey. [16]The Ohio State University Division of Human Genetics, Department of Internal Medicine, 460 W 12th Ave, Columbus, Ohio 43210, USA. [17]Department of Molecular and Human Genetics, Baylor College of Medicine, Houston, TX 77030, USA. [18]Human Genome Sequencing Center, Baylor College of Medicine, Houston, TX 77030, USA. [19]Department of Pediatrics, Baylor College of Medicine, Houston, TX 77030, USA. [20]Texas Children's Hospital, Houston, TX 77030, USA. [21]Institute of Human Genetics, Technische Universität München, München 81675, Germany. [22]Luxembourg Center for Systems Biomedicine, University Luxembourg, Esch-sur-Alzette 4365, Luxembourg. [23]Center for Rare Childhood Disorders, The Translational Genomics Research Institute, Phoenix, AZ 85004, USA. [24]Department of Neuropediatrics, Christian-Albrechts-University Kiel and University Hospital Schleswig-Holstein, Campus Kiel 24105, Germany. [25]Division of Clinical Genomics, Ambry Genetics, Aliso Viejo, CA 92656, USA. [26]Pediatric Department B' Emek Medical Center, Afula 1834111, Israel. [27]Rappaport School of Medicine, Technion, Haifa 3200003, Israel.

[28]Northern German Epilepsy Center for Children and Adolescents, Schwentinental-Raisdorf 24223, Germany. [29]Department of Neurology, University of Michigan, Ann Arbor, MI 48109, USA. [30]Present address: Department of Genetics, University of Alabama, Birmingham, AL 35233, USA. [31]Present address: Department of Pharmacology and Pharmacotherapy, University of Pecs, Pecs 7622, Hungary. These authors contributed equally: Aleksandra Siekierska, Hannah Stamberger. These authors jointly supervised this work: Peter de Witte, Peter De Jonghe.

## C4RCD Research Group

Chris Balak[23], Newell Belnap[23], Ana Claasen[23], Amanda Courtright[23], Matt de Both[23], Matthew J. Huentelman[23], Marcus Naymik[23], Ryan Richholt[23], Ashley L. Siniard[23], Szabolcs Szelinger[23], David W. Craig[32] & Isabelle Schrauwen[33]

[32]Department of Translational Genomics, Keck School of Medicine, University of Southern California, Los Angeles, CA 90033, USA. [33]Center for Statistical Genetics, Department of Molecular and Human Genetics, Baylor College of Medicine, One Baylor Plaza 700D, Houston, TX 77030, USA

## AR working group of the EuroEPINOMICS RES Consortium

Zaid Afawi[34], Rudi Balling[22], Stéphanie Baulac[35,36,37,38,39], Nina Barišić[40], Hande S. Caglayan[41], Dana Craiu[42], Rosa Guerrero-López[43], Renzo Guerrini[44], Helle Hjalgrim[45,46], Johanna Jähn[24], Karl Martin Klein[47], Eric Leguern[35,36,37,38,39], Johannes R. Lemke[48], Holger Lerche[49], Carla Marini[44], Rikke S. Møller[45,46], Hiltrud Muhle[24], Felix Rosenow[47], Jose Serratosa[43], Arvid Suls[2,3,53], Ulrich Stephani[24], Katalin Štěrbová[50], Pasquale Striano[51] & Federico Zara[52]

[34]Department of Physiology and Pharmacology, Tel Aviv University Medical School, Ramat Aviv 69978, Israel. [35]Sorbonne Université, UPMC Univ Paris 06, UMR S 1127, Paris 75013, France. [36]INSERM, U1127, Paris 75013, France. [37]CNRS, UMR 7225, Paris 75013, France. [38]Institut du Cerveau et de la Moelle épinière (ICM), Hôpital Pitié-Salpêtrière, Paris 75013, France. [39]Department of Genetics, Assistance Publique des Hôpitaux de Paris (AP-HP), Hôpital Pitié-Salpêtrière, Paris 75013, France. [40]Department of Paediatrics, Clinical Medical Centre Zagreb, University of Zagreb, Medical School, Zagreb 10000, Croatia. [41]Department of Molecular Biology and Genetics, Bogazici University, Istanbul 34342, Turkey. [42]Department of Clinical Neurosciences and Pediatric Neurology Clinic, "Carol Davila" University of Medicine, Al. Obregia Hospital, Bucharest 050474, Romania. [43]Department of Neurology, Neurology Lab and Epilepsy Unit, IIS-Fundación Jiménez Díaz UAM and CIBERER, Madrid 28040, Spain. [44]Pediatric Neurology, Neurogenetics, and Neurobiology Unit and Laboratories, A. Meyer Children's Hospital, University of Florence, Florence 50139, Italy. [45]Danish Epilepsy Centre, Dianalund 4293, Denmark. [46]Institute for Regional Health research, University of Southern Denmark, Odense 5230, Denmark. [47]Department of Neurology, Epilepsy Center Frankfurt Rhine-Main, Goethe-University, Frankfurt am Main 60323, Germany. [48]Institute of Human Genetics, University of Leipzig Hospitals and Clinics, Leipzig 04103, Germany. [49]Department of Neurology and Epileptology, Hertie Institute for Clinical Brain Research, University of Tübingen, Tübingen 72076, Germany. [50]Child Neurology Department, 2nd Faculty of Medicine, Charles University and University Hospital Motol, Prague 150 06, Czech Republic. [51]Pediatric Neurology and Muscular Diseases Unit, Department of Neurosciences, Rehabilitation, Ophthalmology, Genetics, and Maternal and Child Health, University of Genoa, 'G. Gaslini' Institute, Genoa 16147, Italy. [52]Laboratory of Neurogenetics and Neuroscience, 'G. Gaslini' Institute, Genoa 16147, Italy. [53]Present address: Center of Medical Genetics, University of Antwerp and Antwerp University Hospital, Antwerp 2650, Belgium

