## [Peer Review File · Nature Communications]

Reviewer #1 (Remarks to the Author):

With interest I read the manuscript by Siekierska et al. entitled "Developmental encephalopathy with microcephaly linked to bi-allelic VARS variants is phenotypically recapitulated in a vars knockout zebrafish model".

The authors present 6 families, of which 2 were previously reported without all details, with 9 patients in whom bi-allelic VARS mutations were identified. Additionally the modeled the mutations to provide further evidence for pathogenicity of the variants.

I have a few points that the authors may wish to address to help clarify some questions that arose while reviewing the manuscript.

- How were the families identified? the authors report on 6 families, of which two were identified/reported by Karaca et al. I however wonder what the rationale was to put these cases together. Was this based on phenotypic characterization? GeneMatcher exchange programs?

- when referring to the phenotypes of the patients, please be consistent in whether or not to address 'the exact patient ID' versus 'three patients' as now sometimes details are provided, and sometimes these are not. Similarly, sometimes percentages are provided (for instance 78% of patients have epileptic seizures) whereas in other occasions more subjective terms as 'most patients' is used without this quantification.

- The authors mention 'that parents self reported developmental decline' for patient 3. What milestone(s) did the parents report this on? Can this be substantiated by objective measures in medical reports?

- Regarding variant identification: from family pedigree(s) and history, families 1 and 3 are not necessarily 'recessive'. Hence, authors likely also looked/prioritized for X-linked (family 1) and/or dominant (de novo) variants in Family 3. If so, the authors should include this information and report on the potential findings related hereto. For figure 1b, it might be helpful to indicate which mutations were identified in which patients, and whether they were (comp) heterozygous or homozygous.

- Regarding the protein modeling: The authors report to have used three categories for variant classification. They refer to 'inspection of the corresponding residues' to based this classifications. What did this inspection consist of? and is this a subjective/observant biased inspection, or were 'set criteria' used? It would also be informative to provide the mutations which were classified per category.

- From a genetic point of view, Table 1 contains relevant information, but could also be presented in a supplementary file. For clarity/overview purposes, and the great diversity of different assays performed on the different mutations, QARS, KARS and AARS. Have the authors tried to, or could the authors speculate on the usefulness of, screening a cohort of patients - previously tested negative for mutations in these genes - for mutations in VARS?

- Figure 2: for clarity, please refer to patient or family IDs instead of 1 and 2.

- Figure 3: part b, the yellow line is not visible. It would be my understanding that it is fully coinciding with the WT (gree) line, but is there any way to make it visable? Also, for x-axis, please indicate which type of 'days'. I assume this is dpf?

- Zebrafish vars -/- are incompatible with life. Yet, in humans, functional bi-allelic null-alleles are observed. Can the other speculate on this observation? Is this due to the (potential) redundancy of human ARS genes?

- Figure 4 (and related text in the main body): panel 4a, can the authors explain or provide a hypothesis why the 4 dpf vars +/- is not similar to the wt +/-?

- Figure 5 (and related tekst in the main body): Panel f: there is data and/or comparison to the vars+/-GFP inj control. Why not? ALso, in panels g-i, the vars -/- supplemented with human WT VARS is not included. From these figures alone, it cannot be used that the wt is rescuing the phenotypes measured (e.g. the positive control seems absent).

What is the explanation of the authors, that only for days 4 and 5 post fertilizatoin there is a significant rescue? And similarly, whereas not statistically significant, 7 dpf, the effect seems opposite.

- The VARS protein is an enzyme. The authors have performed so many different (functional) experiments in (a) model organis(s), but one 'obvious' seems missing: can the VARS enzyme activity in patients be measured? And if so, have the authors made attempts to measure this activity?

- Overall, the manuscript is rather lengthy and would benefit to summarize the results section as the figures and text containing redundant information.

Reviewer #2 (Remarks to the Author):

Siekierska and co-workers report evidence functionally linking bi-allelic mutations in valyl-aminoacyl tRNA synthetase (VARS) and severe developmental brain defects in human. Their study presents human genetic data associating bi-allelic VARS missense mutations to severe neurodevelopmental defects, focusing on seven mutations (compound heterozygous and homozygous). One of these has a missense mutation over an early frameshift predicted to be null. One of the missense (p.Gly822Ser) is predicted to prevent t-RNA binding and the authors very elegantly showed that the gene carrying this mutation is unable to rescue the lack of VARS function in yeast, while the other

missense mutations found do, therefore importantly also showing that these other patient mutations only affect partially VARS function. Most of the functional study is then unfortunately focussed on a zebrafish complete null mutant. Complete lack of VARS is obviously predicted to be lethal as translation of most proteins would suffer from the lack of this enzyme so the results obtained in the zebrafish are of limited impact. The most important zebrafish results are the rescue experiments of the null mutant by the various missense mutant transcripts. These rescues require much deeper analysis as they are a unique avenue to understand the impact of the missense modifications found in the patients. Overall, the link between VARS mutations and specific cases of microcephaly is a very important finding of broad interest to Nature comms readership but the study requires more careful functional data before considering the publication further.

Specific comments:

- The importance of the finding that all mutations not predicted to affect the function completely are able to supply enough function in yeast has to be highlighted better.
- Fig. 3d are not sufficient, measures and pictures from 24 and 48hpf are needed too to evaluate progression.
- The zebrafish habituation learning (rather than calling it cognition) test is actually showing habituation in homozygous, in a similar way to siblings. In average, they respond less to light cues with time, starting with a more pronounced response (interesting!). These results need better analysis.
- Epileptiform activity not convincing. Recording of tectal activity in wildtype can also show these bursts. Need to show measurements over time in wildtype, heterozygous and homozygous to convince.
- I am very puzzled by the rescue at day 12 of 30% of the homozygous by a RNA injection made at 1-cell stage. The authors need to show that presence of VARS proteins in extracts of 9 dpf rescue null. If the protein (or RNA) is so stable, is the null doing OK without an essential ARS until 48hpf or so thanks to maternal contribution? If so, why the maternal contribution is not as stable as the RNA injected at 1-cell stage?
- How do the author measure null rescued behaviour? Injection is done at 1-cell stage in progeny of heterozygous crosses. How do they make sure to measure touch response, head size and behaviour in nulls? Where is the data for the siblings?
- Rescue experiments with the missense RNAs are the most meaningful experiments to link these to disorders. These need to be done with outmost care and careful quantification. They need to show some brain morphology and neuronal stainings quantified between siblings and mutants.

Reviewer #3 (Remarks to the Author):

The curious thing about the zebrafish VARS phenotype is the slow onset. Although the early development of the brain (up to day 3) is not analysed, there is an obvious deterioration in structure and function from 5 days. What do the authors think is going on in early development? Is VARS not important early? But authors suggest it is expressed early? We really need more accurate expression data for VARS in embryonic and larval development. The reference to the online expression data is too vague and not good enough. Where and when is it expressed in the brain? Could expression profile help explain the late onset phenotype?

The morphological analysis of zebrafish larvae in Figure 3d and e is insufficient. At what time in development do the malformations occur? We need to see structures at earlier timepoints. Also the single sections given for the day 6 larvae in Figure 3e aren't really sufficient to tell whether equivalent sections are being compared or how widespread the problems are in the brain. Is the gross loss of structure we see at 5 and 6 days due to cell death or are the structures reduced from the outset? In this regard, is the reduced brain size in the zebrafish model really equivalent to human microcephaly, or is the fish phenotype driven by relatively late cell death? The "fitness" (Figure 3c) of the larvae begins to decrease from day 4 and survival decreases from day 8, so maybe the whole animal is just slowly dying from day 4? That wouldn't make it a very good model I suspect. I think an analysis of cell death would be informative from days 1 through to 6. I also think to have confidence that the zebrafish is telling us something relevant to human patients it would be good to give more details on the human microcephalies. At present there is just a very vague statement about the human patient microcephalies (bottom page 7).

For the experiments involving mRNA injections to test whether they can rescue the zebrafish phenotypes, how long do the authors think the injected RNA lasts in the embryo/larvae? Is it still present and active at 3 days and beyond?

Reviewer #1

With interest I read the manuscript by Siekierska et al. entitled "Developmental encephalopathy with microcephaly linked to bi-allelic VARS variants is phenotypically recapitulated in a vars knockout zebrafish model".

The authors present 6 families, of which 2 were previously reported without all details, with 9 patients in whom bi-allelic VARS mutations were identified. Additionally, the modeled the mutations to provide further evidence for pathogenicity of the variants. I have a few points that the authors may wish to address to help clarify some questions that arose while reviewing the manuscript.

Response: We thank Reviewer #1 for these constructive comments and we would also like to inform him/her that in the revised version of our manuscript we have included one additional family (family VII) with the exact same missense substitution as family VI.

1. *How were the families identified? the authors report on 6 families, of which two were identified/reported by Karaca et al. I however wonder what the rationale was to put these cases together. Was this based on phenotypic characterization? GeneMatcher exchange programs?*

Response: The initial family was identified through the EuroCores program EuroEPINOMICS. Families II and new family VII were diagnosed in diagnostic laboratories and included through international collaborations and families III and VI were ascertained through the program GeneMatcher as stated in the methods section. We have included a paragraph in the results section to clarify this as well.

2. *When referring to the phenotypes of the patients, please be consistent in whether or not to address 'the exact patient ID' versus 'three patients' as now sometimes details are provided, ad sometimes these are not. Similarly, sometimes percentages are provided (for instance 78% of patients have epileptic seizures) whereas in other occasions more subjective terms as 'most patients' is used without this quantification.*

Response: We thank the reviewer for this observation and have adapted the respective paragraph to make this more consistent.

3. *The authors mention 'that parents self reported developmental decline' for patient 3. What milestone(s) did the parents report this on? Can this be substantiated by objective measures in medical reports?*

Response: The mother of patient 3 observed at the age of 9 months that the patient

had been able to hold his head up and used to push up with his arms while prone before, but would not do this anymore. This was documented as such in the medical reports but there were no objective measures indicating regression. We have decided to omit this part in the manuscript as this is indeed a subjective observation made by the parents; we only mentioned this in the Supplementary Table 1.

4. *Regarding variant identification: from family pedigree(s) and history, families 1 and 3 are not necessarily 'recessive'. Hence, authors likely also looked/prioritized for X-linked (family 1) and/or dominant (de novo) variants in Family 3. If so, the authors should include this information and report on the potential findings related hereto. For figure 1b, it might be helpful to indicate which mutations were identified in which patients, are whether they were (comp) heterozygous or homozygous.*

Response:

- WGS and WES analysis for family I and III was indeed also performed under a *de novo* dominant and/or X-linked model. Details of the analysis pipeline for all families have been included in the supplementary notes where we have also specified variant prioritisation based on different hypothesized inheritance.
- We have included all mutations in the pedigrees of Fig. 1a so it is clear which patient had which compound heterozygous or homozygous mutation based on this figure.

5. *Regarding the protein modeling: The authors report to have used three categories for variant classification. They refer to 'inspection of the corresponding residues' to based this classifications. What did this inspection consist of? and is this a subjective/observant biased inspection, or were 'set criteria' used? It would also be informative to provide the mutations which were classified per category.*

Response: To address the reviewers concern and provide a presentation format that is consistent across both papers (this one and the Friedman et al manuscript), we extended the modeling study to better examine the likely structural consequences of the individual mutants. In the revised manuscript and Supplementary Table 3, we present an explicit structural comparison between the structures of the T. thermophilus VARS complex with the native residues at the substituted positions, and the altered residues found in the mutants. Potential contacts (hydrogen bonds, salt bridges and hydrophobic contacts) to nearby residues were examined in both cases, and then the results were displayed using Pymol (Fig. 1c and d). Employing this analysis, the mutants fall into two (and not three) different categories. The first category (6/7) is composed of those substitutions that are likely to have a direct or indirect effect on protein structure, owing to a loss of a stabilizing with one or more nearby residues. The specific structural consequences of the individual mutants are described in the supplementary notes. The second category is defined by the mutant substitution (p.Arg942Gln), which alters a direct contact to the transfer RNA substrate. Finally, the p.Leu888Phe variant is located in an insertion sequence that is missing in T. thermophilus enzyme and therefore cannot be explicitly modeled. However, the insertion is located proximal to the tRNA anticodon, which suggests the potential for a tRNA binding effect. This classification scheme is based on direct analysis of the T. thermophilus structure, without making arbitrary distinctions about mutant classes. As noted in the revised text, we conclude that the most likely explanation for the effect of the

mutant substitutions is that they destabilize the protein, decreasing their intracellular levels and producing a loss of aminoacylation function. The text and supplementary notes have been modified to indicate these conclusions.

6. *From a genetic point of view, Table 1 contains relevant information, but could also be presented in a supplementary file. For clarity/overview purposes, and the great diversity of different assays performed on the different mutations, QARS, KARS and AARS. Have the authors tried to, or could the authors speculate on the usefulness of, screening a cohort of patients - previously tested negative for mutations in these genes - for mutations in VARS?*

Response:

- We thank the reviewer for this suggestion. We have moved Table 1 to the supplementary file as a Supplementary Table 2.
- We agree with the reviewer that it would be useful to specifically screen for VARS mutations in a cohort of patients with developmental delay, seizures and microcephaly, phenotypic features that significantly overlap with the phenotype associated with other proteins involved in translation, like AARS, QARS and KARS, to see what the specific mutation yield is. We did however not have access to previously published (and often very heterogeneous) patients cohorts tested for these three genes, nor an own cohort (with this specific phenotype) large enough to say something on gene specific phenotypic features.

In clinical practice, and given the rarity of mutations in ARS genes, patients/families will likely be identified through NGS approaches such as WES that include all ARS genes or other genes involved in protein translation at once. As such, when confronted with a patient with the above mentioned phenotype, specific attention should be given to this set of genes during analysis. We have added a comment in the discussion.

7. *Figure 2: for clarity, please refer to patient or family IDs instead of 1 and 2.*

Response: We thank the reviewer for noting this. We agree this is unclear and have changed “1” and “2” into “patient 4” and “patient 5” respectively.

8. *Figure 3: part b, the yellow line is not visible. It would be my understanding that it is fully coinciding with the WT (green) line, but is there any way to make it visible? Also, for x-axis, please indicate which type of 'days'. I assume this is dpf?*

Response: The figure has been adjusted to the reviewer's suggestions by making both orange and green lines visible in the graph in Figure 3c and changing the x-axis title to “dpf” in Figures 3c and 3d.

9. *Zebrafish vars -/- are incompatible with life. Yet, in humans, functional bi-allelic null-alleles are observed. Can the other speculate on this observation? Is this due to the (potential) redundancy of human ARS genes?*

Response: As far as we are aware no bi-allelic VARS null variants have been observed in humans. In public databases ExAC and gnomAD null variants have only be observed in heterozygous state. We speculate complete loss of function is not viable, nor in humans, nor in zebrafish. This is discussed in more detail in the manuscript discussion.

10. *Figure 4 (and related text in the main body): panel 4a, can the authors explain or provide a hypothesis why the 4 dpf vars +/- is not similar to the wt +/+?*

Response: We have also noticed a difference in the locomotor activity between *vars +/-* and *vars +/+* at 4 dpf. Proper wiring of neurons within circuits early in development is essential for subsequent behavior (Sternberg et al, 2015). Our hypothesis is that since at 4 dpf zebrafish brain is not yet fully developed, there might be a partially functional neuronal wiring present, together with a relative lack of well-coordinated firing, which make the behavioral data somewhat unreliable at this stage. This is also consistent with previous reports, which state that there is more variability in the locomotor activity for younger (rather than older) larvae (Ingebretson et al, 2013). Moreover, the larvae hatch spontaneously between 2 and 3 dpf, therefore in line with this we have excluded the locomotor data for 3 dpf larvae (also from Figure 5e).

11. *Figure 5 (and related text in the main body): Panel f: there is data and/or comparison to the vars+/-GFP inj control. Why not? Also, in panels g-i, the vars -/- supplemented with human WT VARS is not included. From these figures alone, it cannot be used that the wt is rescuing the phenotypes measured (e.g. the positive control seems absent). What is the explanation of the authors, that only for days 4 and 5 post fertilization there is a significant rescue? And similarly, whereas not statistically significant, 7 dpf, the effect seems opposite.*

Response: We thank the reviewer for this comment.

a. *vars+/-* GFP-injected controls have been added to panel e in Figure 5, according to the reviewer's remark.

b. Results for *vars-/-* supplemented with human WT VARS are presented separately in panels a-c in Figure 5, where the rescue effect is clearly visible. Panels g-i show that mutated VARS cannot rescue the microcephaly phenotype, similarly to *vars-/-* ctrl-injected larvae. We deliberately chose not to include *vars-/-* WT VARS injected larvae into panels g-i. We believe that adding these data points will render the graphs too crowded and therefore difficult to interpret, considering that already two controls are included, i.e. positive, *vars+/-* ctrl-injected, and negative, *vars-/-* ctrl-injected. Moreover, adding these data would be a repetition of data present in panels a-c.

c. Concerning the rescue of swimming activity, at 4 and 5 dpf we assume the influence of WT VARS mRNA, which at 6 and 7 dpf is not effective anymore. In order to confirm this, we have performed qPCR to detect WT VARS mRNA in the injected *vars -/-* embryos at 1, 3 and 5 dpf (was added as Figure 5f). When comparing to 1 dpf, at 3 and 5 dpf there was 24.55% and 10.42% WT VARS mRNA left, respectively, which confirms our findings. We strongly believe that at 7 dpf, there would be no mRNA present, therefore the rescue effect is gone.

12. *The VARS protein is an enzyme. The authors have performed so many different (functional) experiments in (a) model organism(s), but one 'obvious' seems missing: can the VARS enzyme activity in patients be measured? And if so, have the authors made attempts to measure this activity?*

Response: We appreciate the reviewer's suggestion, and we have measured the aminoacylation activity in several patient cells lines that we were able to gain access to. These include fibroblasts from two siblings with the compound heterozygous mutation (p.Leu78Argfs*35 /p.Arg942Gln), and lymphoblasts from

family 1 and 6 with the p.Leu434Val/p.Gly822Ser and the homozygous p.Arg404Trp variant respectively. Extracts from these cells were prepared, as well as extracts from control ATCC fibroblasts and lymphoblasts. The aminoacylation reactions included total human placental tRNA as the tRNA source, and all extracts were assayed for VARS activity and TARS activity (as control). Aminoacylation activity was determined as a specific activity (pmoles valyl-tRNA^{Val} formed/min/ng protein). As seen in revised Figure 2d and f, the results indicate that, without exception, the subjects with either the compound heterozygous mutation or the homozygous mutation exhibited substantially reduced valylation activity, but normal levels of threonyl-tRNA synthetase. The manuscript has been updated with this information.

13. *Overall, the manuscript is rather lengthy and would benefit to summarize the results section as the figures and text containing redundant information.*

Response: We thank the reviewer for this comment and have tried to make the manuscript (and tables) more concise, omitting redundant information.

Reviewer #2

Siekierska and co-workers report evidence functionally linking bi-allelic mutations in valyl-aminoacyl tRNA synthetase (VARS) and severe developmental brain defects in human. Their study presents human genetic data associating bi-allelic VARS missense mutations to severe neurodevelopmental defects, focusing on seven mutations (compound heterozygous and homozygous). One of these has a missense mutation over an early frameshift predicted to be null. One of the missense (p.Gly822Ser) is predicted to prevent t-RNA binding and the authors very elegantly showed that the gene carrying this mutation is unable to rescue the lack of VARS function in yeast, while the other missense mutations found do, therefore importantly also showing that these other patient mutations only affect partially VARS function.

Most of the functional study is then unfortunately focussed on a zebrafish complete null mutant. Complete lack of VARS is obviously predicted to be lethal as translation of most proteins would suffer from the lack of this enzyme so the results obtained in the zebrafish are of limited impact. The most important zebrafish results are the rescue experiments of the null mutant by the various missense mutant transcripts. These rescues require much deeper analysis as they are a unique avenue to understand the impact of the missense modifications found in the patients.

Overall, the link between VARS mutations and specific cases of microcephaly is a very important finding of broad interest to Nature comms readership but the study requires more careful functional data before considering the publication further.

Response: We thank Reviewer #2 for positive comments and for careful inspection of the zebrafish data. We have taken reviewer's comments thoroughly in consideration and tried to address most of them according to the suggestions made.

Specific comments:

1. *The importance of the finding that all mutations not predicted to affect the function completely are able to supply enough function in yeast has to be highlighted better.*

Response: We thank the reviewer for this comment and agree that this is an

important point to highlight in the discussion. Specifically, homozygosity or compound heterozygosity for null *VARS* alleles would be lethal. Therefore, it is not surprising that some of the disease-associated *VARS* variants retain function in our yeast model. Furthermore, our yeast assay is unlikely to have the resolution to detect more subtle alterations in *VARS* function, which would be an expected consequence of a hypomorphic allele. To describe this issue in more detail, we have added text to the discussion on page 19.

2. *Fig. 3d are not sufficient, measures and pictures from 24 and 48hpf are needed too to evaluate progression.*

Response: We have added additional images of 1 and 3 dpf larvae showing the progression of the phenotype (Fig.3e and Supplementary Fig. 5a). The morphological changes could be observed from 3 dpf onwards. Furthermore, a detailed histological analysis from 1 to 5 dpf was performed (Fig. 3f and Supplementary Fig. 5b) that demonstrated very subtle changes at 2 dpf in the brains of *vars* *-/-* embryos that progressed over time, resulting in disrupted brain architecture, reduced jaw structures, delayed retinal lamination, reduced lens and corneal edema at 5 dpf.

3. *The zebrafish habituation learning (rather than calling it cognition) test is actually showing habituation in homozygous, in a similar way to siblings. In average, they respond less to light cues with time, starting with a more pronounced response (interesting!). These results need better analysis.*

Response: We are thankful to the reviewer for this relevant comment and the suggestion to provide a better interpretation of the habituation assay.

We agree with the reviewer that it is more accurate to call this type of experiment a habituation test rather than a cognitive assay, because the latter term refers to more complex brain processes such as e.g. learning and memory, decision-making, context-specific cognitive judgments, which we did not study in 6 dpf larvae. This has been adjusted in the manuscript.

We re-analyzed the outcome of the habituation assay by fitting straight lines through all the data points for each genotype group. Indeed, as the reviewer correctly pointed out, *vars* *-/-* larvae could adapt to dark flashes (DFs) similarly to its siblings, as the movement values decreased over time and there was no statistical difference between the slopes ($p=0.0658$). Interestingly however, the motion of *vars* knockout larvae in response to DF was significantly increased ($p<0.0001$). This hyperactivity suggests a neuromodulatory effect, as the behavior depends on sensory input and is regulated by neuronal connectivity of the hindbrain and spinal cord and multiple neurotransmitter pathways (Copmans et al, 2016). We also speculate that the increase in movement could be due to the epileptiform activity occurring at 6 dpf in *vars* *-/-* larvae.

We have changed the text in the manuscript accordingly, as well as modified the Figure 4b by adding the linear slopes fitted to the movement points and a panel legend below the graph, so that the readers can now better understand the outcome and design of the habituation experiment.

4. *Epileptiform activity not convincing. Recording of tectal activity in wildtype can also show these bursts. Need to show measurements over time in wildtype, heterozygous and homozygous to convince.*

Response: We performed additional local field potential recordings at 5, 6 and 7 dpf larvae in order to demonstrate the specificity of the observed epileptiform activity. These results summarized in Figure 4d show that during 5-7 dpf the *vars*^{-/-} larvae had significantly more events than *vars*^{+/-} or *vars*^{+/+}. Abnormal brain activity was observed in 68.57% (24/35) 5 dpf *vars*^{-/-}, 5.26% (1/19) 5 dpf *vars*^{+/+}, and 0% (0/31) 5 dpf *vars*^{+/-}; 47.62% (23/48) 6 dpf *vars*^{-/-}, 5% (1/20) 6 dpf *vars*^{+/+}, 6.67% (2/30) *vars*^{+/-}; and 52.63% (10/19) 7 dpf *vars*^{-/-}, 8.33% (1/12) 7 dpf *vars*^{+/+} and 0% (0/13) 7 dpf *vars*^{+/-}.

We would also like to mention that some of the observed events in the recorded larvae (both in *vars*^{-/-} and siblings) might resulted from movement or artifacts, commonly occurring in these types of recordings.

5. *I am very puzzled by the rescue at day 12 of 30% of the homozygous by a RNA injection made at 1-cell stage. The authors need to show that presence of VARS proteins in extracts of 9 dpf rescue null. If the protein (or RNA) is so stable, is the null doing OK without an essential ARS until 48hpf or so thanks to maternal contribution? If so, why the maternal contribution is not as stable as the RNA injected at 1-cell stage?*

Response: We thank the reviewer for this comment.

a. The difference in the survival that the reviewer pointed out is not statistically significant, so we cannot claim that WT VARS prolonged the survival of *vars* knockout larvae. Therefore, we removed this graph from the figure in order to avoid confusion. Nevertheless, we believe that *vars* mRNA is not present at 9 dpf in rescued larvae, but do not know the effect of *vars* on the turnover of its interacting partners/downstream effectors, all of which could play a role in determining the null phenotype.

b. We agree with the reviewer that the lack of null phenotype observed until 48 hpf is most probably influenced by maternal *vars* expression. In order to find this out we performed qPCR on wild type AB embryos at 2 hpf (~64-cell stage), so before the maternal zygotic transition (MZT) occurs in zebrafish. Detected *vars* mRNA at this stage, expressed from maternal genes, was comparable to *vars* expression at 24 hpf (figure below), confirming the maternal contribution for the lack of early phenotype of *vars*^{-/-} larvae.

c. During MZT, maternal mRNA is rapidly cleared by maternally supplied factors and newly synthesized zygotic gene products (Walser et al., 2011). When looking at the expression of WT *vars* in 24 hpf *vars*^{-/-} embryos, no or only trace amounts of mRNA could be detected (qPCR performed at 24 hpf embryos using primers that bind to the sequence within *vars* 4 bp deletion; data not shown).

In order to estimate the time maternal *vars* is present in *vars*^{-/-} larvae (there are no zebrafish antibodies available to verify the protein level and human/mouse antibodies do not cross-react with zebrafish protein), we determined the half-life of its human homologue in a fibroblast cell line by cyclohexamide treatment. Considering the fact that zebrafish and human VARS are highly conserved and are essential enzymes for proper development, we hypothesized that their half-lives would be similar. The results (shown below) demonstrated that human VARS had relatively long half-life, ~48 hours. When extrapolated to zebrafish, this outcome could explain why during the first 48 hpf no obvious phenotype could be seen in *vars* knockout larvae.

6. *How do the author measure null rescued behaviour? Injection is done at 1-cell stage in progeny of heterozygous crosses. How do they make sure to measure touch response, head size and behaviour in nulls? Where is the data for the siblings?*

Response: During rescue experiments each injected larva was monitored separately and its touch response, behaviour and head size were measured. After every experiment, the larvae were lysed and sequenced (as described in materials and methods) in order to determine the genotype to correctly interpret obtained results. The data for *vars*^{+/+} GFP-injected larvae were added to the locomotor activity graph in Figure 5e. We have all the data for *vars*^{+/-} siblings, however since these larvae morphologically and physiologically resemble their WT siblings, they were not shown in the graphs for the sake of clarity. In our opinion, they do not provide any additional information that could be helpful to interpret or understand the current data from the rescue studies.

7. *Rescue experiments with the missense RNAs are the most meaningful experiments to link these to disorders. These need to be done with outmost care and careful quantification. They need to show some brain morphology and neuronal staining quantified between siblings and mutants.*

Response:

a. We are thankful for emphasizing the importance of rescue experiments. We have reviewed all the rescue data and adjusted Figure 5 by adding control larvae to the tracking experiment (panel e) and removing the survival experiment since the comparison between *vars*^{-/-} and wild type siblings was not statistically significant. Moreover, we performed qPCR experiments in order to determine the levels of supplied human WT VARS mRNA at 1, 3 and 5 dpf in injected *vars*^{-/-} larvae and added them as Figure 5f.

b. Since the histology experiments aiming to investigate more in detail *vars*^{-/-} phenotype did not show apparent changes in the brain morphology at 1 and 2 dpf,

we did not consider to use this laborious technique to evaluate the rescue at later days, especially since at 3 dpf we had already provided partial rescue data from head/brain/eye measurements.

Regarding the staining of neuronal markers, even though we agree with the reviewer that the outcome would be informative, we believe that these experiments are beyond the scope of this manuscript due to time constraints.

Reviewer #3 (Remarks to the Author):

Response: We thank Reviewer #3 for the detailed comments on the zebrafish model.

1. *The curious thing about the zebrafish VARS phenotype is the slow onset. Although the early development of the brain (up to day 3) is not analysed, there is an obvious deterioration in structure and function from 5 days. What do the authors think is going on in early development? Is VARS not important early? But authors suggest it is expressed early? We really need more accurate expression data for VARS in embryonic and larval development. The reference to the online expression data is too vague and not good enough. Where and when is it expressed in the brain? Could expression profile help explain the late onset phenotype?*

Response:

a. We performed whole *in situ* hybridization (WISH) experiments in order to examine *vars* expression at different embryonic and larval stages in whole zebrafish embryo. Selected images were added to the main manuscript as panel a in Figure 3, whereas all the stained stages were provided as Supplementary Figure 3. Although not detectable during gastrulation stage (at 6 hpf), *vars* mRNA was found to be ubiquitously expressed at 18-somite stage at 18 hpf, with more distinctive expression in the brain region and in the prospective eye as well as in the hematopoietic intermediate cell mass and somites, which was maintained till 24 hpf. From 36 hpf the expression of *vars* became restricted to the developing brain, highly enriched in the midbrain, midbrain-hindbrain boundary, and hindbrain. After 48 hpf *vars* expression was also observed in other developing organs including branchial arches, liver, pancreas, and intestine (Fig. 3a and Supplementary Fig. 3). These dynamic expression patterns strongly suggest an essential role of *vars* in the midbrain/hindbrain development, while the expression outside CNS also suggests multiple roles of *vars* during organogenesis.

b. The slow onset of the zebrafish *vars* phenotype is most probably influenced by maternal *vars* expression. In order to determine the presence of maternal *vars* mRNA, we performed qPCR on wild type AB embryos at 2 hpf, so before maternal zygotic transition occurs. Detected *vars* mRNA at this stage, expressed from maternal genes, was comparable to *vars* expression at 24 hpf (figure below), confirming the maternal contribution for the lack of early phenotype of *vars*^{-/-} larvae.

2. *The morphological analysis of zebrafish larvae in Figure 3d and e is insufficient. At what time in development do the malformations occur? We need to see structures at earlier timepoints.*

Response: Additional images of the phenotype showing the progression have been added to the figure (Figure 3e and Supplementary Figure 5a). Morphological changes could be observed from 3 dpf onwards. Moreover, a histological analysis from 1 to 5 dpf was performed (Figure 3f and Supplementary Figure 5b) in order to determine in detail the onset and progression of the malformations observed in *vars*^{-/-} larvae. Apparent changes could be observed from 3 dpf, where *vars*^{-/-} larvae displayed reduced tissue mass and abnormal organization of the midbrain as well as disruption of the retinal pigment epithelium (RPE), that progressed over time resulting at 5 dpf in disrupted brain architecture, reduced jaw structures, delayed retinal lamination, reduced lens and corneal edema.

For the sake of consistency of Figure 3, we have removed histological data from 6 dpf and put 5 dpf instead, since both were comparable.

3. *Also the single sections given for the day 6 larvae in Figure 3e aren't really sufficient to tell whether equivalent sections are being compared or how widespread the problems are in the brain. Is the gross loss of structure we see at 5 and 6 days due to cell death or are the structures reduced from the outset? In this regard, is the reduced brain size in the zebrafish model really equivalent to human microcephaly, or is the fish phenotype driven by relatively late cell death? The "fitness" (Figure 3c) of the larvae begins to decrease from day 4 and survival decreases from day 8, so maybe the whole animal is just slowly dying from day 4? That wouldn't make it a very good model I suspect. I think an analysis of cell death would be informative from days 1 through to 6.*

Response: We thank the reviewer for these valid comments.

a. As explained in previous comment, more detailed analysis of the brain morphology during the first 5 days of development has been performed. This time only the equivalent posterior forebrain regions were compared by choosing sections where the optic nerve was present.

b. As demanded by the reviewer, we performed the analysis of cell death by staining the active form of caspase-3 on brain sections from 1-5 dpf embryos and larvae. There was no apparent difference in staining between *vars*^{-/-} and its siblings in all days tested in the posterior forebrain structures (Supplementary Fig. 6). These results indicate that the disruption of the brain architecture is not due to cell death but is occurs progressively from the outset (as can be also seen in Figure 3g), which correlates with what has been observed in patients with *VARS* mutations. Our findings indicate that the reduced brain size in *vars* knockout larvae is really

equivalent to human microcephaly, confirming that it is indeed a good model for the human disease.

Of note, the apoptosis observed in *vars*^{+/+} embryos and larvae is a naturally occurring process during development (Cole et al, 2001).

4. *I also think to have confidence that the zebrafish is telling us something relevant to human patients it would be good to give more details on the human microcephalies. At present there is just a very vague statement about the human patient microcephalies (bottom page 7).*

Response: Details of biometry at birth and last follow-up are provided in the supplementary clinical table for all patients. We have specified this part in the results section saying that all patient have progressive microcephaly. After correcting for GA, only two patients have clear congenital microcephaly (HCC < -2SD mean for age), and 1 additional patient already shows a tendency to it. Next to microcephaly most patients have a general failure to thrive with also weight and length > -2 SD below the mean for age.

5. *For the experiments involving mRNA injections to test whether they can rescue the zebrafish phenotypes, how long do the authors think the injected RNA lasts in the embryo/larvae? Is it still present and active at 3days and beyond?*

Response: In order to respond to the reviewer's comment, we performed qPCR on WT *VARS* injected *vars*^{-/-} larvae at 1, 3 and 5 dpf to detect the presence of supplied human WT *VARS* mRNA. These results demonstrated that at 3 and 5 dpf there was still 24.55% and 10.42% WT *VARS* mRNA left, respectively, in comparison to 1 dpf and were added as panel f to Figure 5.

Furthermore, to estimate how long *VARS* protein is present in *vars*^{-/-} injected larvae, we determined *VARS* half-life by performing cycloheximide (CHX) treatment in human fibroblasts (CHX blocks *de novo* protein synthesis). The fibroblasts were treated for 3, 6, 12, 24 and 48 hours with 150 µg/mL CHX and equal numbers of cells were collected. Western blot was performed on all lysates, using an antibody against human *VARS* protein (that does not cross-react with its zebrafish homologue), and the band signal normalized to a house keeping gene was quantified. The results (shown below) demonstrate that human *VARS* has relatively long half-life, ~48 hours, suggesting that it could be present and active beyond 3 dpf.

Reviewer #1 (Remarks to the Author):

I wish to thank the authors for the revised version of their manuscript. They have made great efforts to include all suggestions made by the reviewers. The additional data provided in the revised version is of high quality and substantiates their conclusions made. I have no further questions/remarks for the authors.

Reviewer #2 (Remarks to the Author):

I am happy with most of the responses to my comments. However, the analysis and representation of the histological study done on the zebrafish mutant from day 1 to 5 is not satisfactory. The sections showed are not done at exactly the same angle and more importantly not comparing the same antero-posterior brain areas. The authors need to show, at least in suppl. fig. 5 a succession of AP levels so that we can assess the extent of the differences in some semi-accurate way (or provide para-sagittal sections). Although the authors find the defect at day 2 "mild", the one section shown suggests a rather substantial change in brain organisation. So, please provide a series of sections for the three genotypes (+/+, +/- and -/-) and word your conclusion more rigorously. A requirement during the second day of development may better explain how a RNA injection manages to rescue so well.

Reviewer #3 (Remarks to the Author):

I'm sorry to say but I really don't trust the zebrafish analyses. It is just not done with sufficient precision, depth or care to be included in this manuscript. The structural analysis of vars mutant brains remains very superficial and the images seem to me to be inconsistent with the authors conclusions. For example the vars-/- brains at 4 and 5 days are clearly smaller than the wt and hets (Fig 3f and h) and I think also smaller than the vars-/- brain sections at day3. So some tissue volume must have been lost in the mutants. But the analysis of cell death is interpreted as showing no difference in cell death between groups. I don't see how this can be the case, and the staining and resolution of the sections used as evidence for cell death (SuppFig 6) is too poor to tell if there is signal and where it is. Is all the brown stain positive signal for cell death in this Figure? Unlikely. Its just too hard to tell.

Furthermore many of the sections in Fig 3F are exactly the same as in SuppFig5 and there is no confidence the sections come from equivalent levels of the brain when we try to compare the various groups. They are roughly equivalent levels but not sufficiently precise to be good for comparisons. Also the tissues tears in 3d vars-/- (red arrows in Fig3f) just look like artifacts from sectioning, not real structural abnormalities.

I think you can conclude from this zebrafish analyses that vars is required for normal teleost brain development, but exactly what the abnormalities are in vars mutants is very uncertain. And whether they are related to human phenotypes is very uncertain. I think the authors will regret including this level of analysis of a fish phenotype in their paper on a human neurological defect. I don't think it adds anything compelling to their main story.

Reviewer #2

I am happy with most of the responses to my comments.

1. However, the analysis and representation of the histological study done on the zebrafish mutant from day 1 to 5 is not satisfactory. The sections showed are not done at exactly the same angle and more importantly not comparing the same antero-posterior brain areas.

Response:

We thank reviewer 2 for carefully looking into our histology experiments and understand the criticism on the sections in the different stadia. We agree that there were several images showing more caudal/ posterior parts of the brain. We therefore carefully revised all our pictures of different sections and replaced those for which we thought quality was insufficient/ the section was not done on the appropriate angle (from 1, 2, 3 and 4 dpf). They are now representing equivalent brain areas (posterior part of the forebrain), allowing us a reliable comparison of the sections. Experts in the field of zebrafish brain histology confirmed our conclusions.

2. The authors need to show, at least in suppl. fig. 5 a succession of AP levels so that we can assess the extent of the differences in some semi-accurate way (or provide para-sagittal sections).

Response:

In order to address the reviewer's comment, we have provided an additional separate supplementary file of series of sections for representative *vars* $-/-$ larvae and its siblings throughout different days, showing succession of AP levels. We believe these series are sufficient to make a reliable comparison between *vars* mutants and siblings, as well as to compare pathology progression over time.

3. Although the authors find the defect at day 2 "mild", the one section shown suggests a rather substantial change in brain organisation. So, please provide a series of sections for the three genotypes (+/+, +/- and -/-) and word your conclusion more rigorously. A requirement during the second day of development may better explain how a RNA injection manages to rescue so well.

Response:

We created a separate supplementary file containing series of sections for representative *vars* $-/-$ larvae and its siblings from 1 to 5 dpf. When comparing again series of sections from 2 dpf, this showed that in *vars* $-/-$ embryos the brain development is impaired; the forebrain is truncated resulting in a substantial size difference, but morphologically it is not entirely changed. We revised the histology part and the conclusions in the manuscript accordingly.

Reviewer #3

I think you can conclude from this zebrafish analyses that vars is required for normal teleost brain development, but exactly what the abnormalities are in vars mutants is very uncertain. And whether they are related to human phenotypes is very uncertain. I think the authors will regret including this level of analysis of a fish phenotype in their paper on a human neurological defect. I don't think it adds anything compelling to their main story.

Response:

We respectfully disagree that the zebrafish data does not add anything compelling to our main story and we regret the reviewer's skepticism concerning the relevance of our zebrafish data. An important part of the zebrafish experiments was to demonstrate that VARS is required for normal brain development, which supports our overall hypothesis that novel loss of function VARS variants cause severe neurodevelopmental phenotypes in humans.

In our manuscript, we describe the generation of a novel zebrafish epilepsy model that recapitulates important features of the human disease. This model is valuable for future study of the pathogenic

mechanisms underlying recessive ARS disease, which could lead to the development of targeted therapies to be tested in *vars* knockout larvae. We also characterized the expression profile of *vars* throughout the development and in rescue experiments demonstrated that three *VARS* mutations were loss of function variants. Furthermore, while some of our data confirm what is already known regarding the phenotype caused by loss of ARS function in zebrafish, our analysis brings novel insights because we are the first to characterize behavioral abnormalities and epileptic activity in *ars* KO zebrafish larvae.

1. I'm sorry to say but I really don't trust the zebrafish analyses. It is just not done with sufficient precision, depth or care to be included in this manuscript. The structural analysis of vars mutant brains remains very superficial and the images seem to me to be inconsistent with the authors conclusions. For example, the vars-/- brains at 4 and 5 days are clearly smaller than the wt and hets (Fig 3f and h) and I think also smaller than the vars-/- brain sections at day3. So some tissue volume must have been lost in the mutants. But the analysis of cell death is interpreted as showing no difference in cell death between groups. I don't see how this can be the case, and the staining and resolution of the sections used as evidence for cell death (SuppFig 6) is too poor to tell if there is signal and where it is. Is all the brown stain positive signal for cell death in this Figure? Unlikely. It's just too hard to tell.

Response:

vars -/- brains were indeed smaller than its +/+ and +/- siblings, but there was no significant difference between brain size in *vars* -/- at 3 and 5 dpf, as shown in Fig. 3h.

We agree that a color comparison of the images with caspase-3 staining is difficult due to abundance of brown signal in the sections and a quantification of the apoptotic cells, enabling more reliable comparison of cell death between the larvae, was missing. Therefore, we have repeated our analysis by counting the number of apoptotic cells (dark brown DAB staining). Three to four equivalent sections were selected for each group and DAB positively stained nuclei were counted using Fiji. The results were expressed as percentage of apoptotic cells of the total cell number within a selected brain area. As the reviewer correctly hypothesized, this quantitative analysis revealed that there is increased cell death from 3 dpf onwards in the brains of *vars* -/- larvae in comparison to its +/+ and +/- siblings.

This suggests that *VARS* is essential for neuronal survival and may correlate to the observation of progressive microcephaly and diffuse atrophy in patients carrying *VARS* mutations. We have adapted the results section in the manuscript accordingly. Though we agree this is an interesting result that should be mentioned in the manuscript, we do not believe this alters our main conclusions.

2. Furthermore, many of the sections in Fig 3F are exactly the same as in SuppFig5 and there is no confidence the sections come from equivalent levels of the brain when we try to compare the various groups. They are roughly equivalent levels but not sufficiently precise to be good for comparisons.

Response:

We thank the reviewer for this comment and carefully revised images from all different sections. We have replaced several images, so that they represent equivalent forebrain levels and are of sufficient quality to reliably compare between various groups. We included in the Fig.3f an overview of sections from 1-5 dpf and added as a supplementary file series of sections from 1-5 dpf.

3. Also, the tissues tears in 3d vars-/- (red arrows in Fig3f) just look like artefacts from sectioning, not real structural abnormalities.

Response:

Although we observed this type of tissue rupture much more frequently in *vars* -/- larvae than in *vars* +/+ and *vars* +/-, we agree it is possible that it might be a result of a fixation process. Therefore,

we have decided to omit it from the manuscript and we have removed the arrow in Fig. 3f drawing attention to this disrupted structure, as we cannot be 100% certain this is not an artefact.

Reviewer #2 (Remarks to the Author):

In response to my point 3, the authors state "When comparing again series of sections from 2 dpf, this showed that in vars-/- embryos the brain development is impaired; the forebrain is truncated resulting in a substantial size difference, but morphologically it is not entirely changed." The sections given as supplementary data are very useful (and need to be added in suppl. data in the manuscript) and show sick/dead cells "flaking" out into the ventricular space in not only the forebrain but also the midbrain so substantial cellular issues arise already at day 2. Please further clarify the text in the main manuscript. For the rest, I am satisfied with the modifications.

Dear Dr. Trenkmann and members of the editorial board,

We thank reviewer 2 for his final comment on our re-revised manuscript "Biallelic *VARs* variants cause developmental encephalopathy with microcephaly that is recapitulated in *vars* knockout zebrafish" (NCOMMS-17-33702C). Please find below a reply to the comment of reviewer 2.

Sincerely

Corresponding authors

Prof. Peter De Jonghe

Prof. Peter de Witte

REVIEWERS' COMMENTS:

Reviewer #2 (Remarks to the Author):

In response to my point 3, the authors state "When comparing again series of sections from 2 dpf, this showed that in vars-/- embryos the brain development is impaired; the forebrain is truncated resulting in a substantial size difference, but morphologically it is not entirely changed." The sections given as supplementary data are very useful (and need to be added in suppl. data in the manuscript) and show sick/dead cells "flaking" out into the ventricular space in not only the forebrain but also the midbrain so substantial cellular issues arise already at day 2. Please further clarify the text in the main manuscript. For the rest, I am satisfied with the modifications.

Reply: According to the suggestion of the reviewer, a sentence describing substantial cellular changes occurring at 2 dpf was added to the main manuscript. Also, the series of sections were added as a supplementary data to the manuscript.